# The AI Hippocampus:
# How Far are We From Human Memory?

**Zixia Jia**[1*], **Jiaqi Li**[1*], **Yipeng Kang**[1*], **Yuxuan Wang**[1*], **Tong Wu**[1], **Quansen Wang**[1,2],
**Xiaobo Wang**[1], **Shuyi Zhang**[1], **Junzhe Shen**[1], **Qing Li**[1], **Siyuan Qi**[1], **Yitao Liang**[2], **Di He**[2],
**Zilong Zheng**[1✉], **Song-Chun Zhu**
*State Key Laboratory of General Artificial Intelligence, BIGAI    Peking University*

**Reviewed on OpenReview:** *https://openreview.net/forum?id=Sk7pwmLuAY*

## Abstract

Memory plays a foundational role in augmenting the reasoning, adaptability, and contextual fidelity of modern Large Language Models (LLMs) and Multi-Modal LLMs (MLLMs). As these models transition from static predictors to interactive systems capable of continual learning and personalized inference, the incorporation of memory mechanisms has emerged as a central theme in their architectural and functional evolution. This survey presents a comprehensive and structured synthesis of memory in LLMs and MLLMs, organizing the literature into a cohesive taxonomy comprising implicit, explicit, and agentic memory paradigms. Specifically, the survey delineates three primary memory frameworks. *Implicit memory* refers to the knowledge embedded within the internal parameters of pre-trained transformers, encompassing their capacity for memorization, associative retrieval, and contextual reasoning. Recent work has explored methods to interpret, manipulate, and reconfigure this latent memory. *Explicit memory* involves external storage and retrieval components designed to augment model outputs with dynamic, queryable knowledge representations—such as textual corpora, dense vectors, and graph-based structures—thereby enabling scalable and updatable interaction with information sources. *Agentic memory* introduces persistent, temporally extended memory structures within autonomous agents, facilitating long-term planning, self-consistency, and collaborative behavior in multi-agent systems, with relevance to embodied and interactive AI. Extending beyond text, the survey examines the integration of memory within multi-modal settings, where coherence across vision, language, audio, and action modalities is essential. Key architectural advances, benchmark tasks, and open challenges are discussed, including issues related to memory capacity, alignment, factual consistency, and cross-system interoperability. By charting the current landscape and identifying critical research directions, this survey aims to inform the development of memory-augmented (M)LLMs that are more flexible, context-sensitive, and aligned with the requirements of real-world intelligent systems. The survey's website is available at https://github.com/bigai-nlco/LLM-Memory-Survey.

---

*Core contributors. ✉ Corresponding to Zilong Zheng.

## Contents

> *"Memory is the treasury and guardian of all things."*
>
> — *Marcus Tullius Cicero, De Oratore*

## 1 Introduction

Recent advancements in artificial intelligence (AI) have led to the development of sophisticated systems, notably (Multi-Modal) Large Language Models (LLMs), which exhibit remarkable capabilities across various domains, from natural language processing and artificial intelligence to software engineering and social sciences (Brown et al., 2020; Hoffmann et al., 2022; Kaplan et al., 2020). Their proficiency extends to tasks such as multi-step reasoning (Wei et al., 2022; Li et al., 2025a) and cross-task generalization (Chatterjee et al., 2024), showcasing their potential to revolutionize various applications. The continuous development of LLMs is a crucial step towards achieving Artificial General Intelligence (AGI), necessitating the incorporation of advanced features that enable these models to autonomously explore and learn from real-world environments.

A pivotal aspect of this development is the integration of memory modules within LLMs, which play a fundamental role in how an agent accumulates knowledge, processes historical experiences, and retrieves information to inform decision-making and actions. By embedding memory capabilities, LLMs can evolve from static entities that rely solely on pre-trained knowledge to dynamic agents capable of continuous learning and adaptation. This transformation allows models to retain and leverage past interactions, thereby enhancing their performance in complex tasks that necessitate long-term planning and a deep contextual understanding. For example, a personal assistant agent with memory capabilities can remember user preferences and previous interactions, thereby delivering more personalized and contextually appropriate responses. Similarly, a trip-planning agent can track user itineraries and preferences, optimizing the efficiency and accuracy of its recommendations. In the healthcare sector, memory-enabled models can maintain comprehensive patient histories, leading to more precise diagnoses and tailored treatment plans. In educational settings, such models can monitor student progress and customize educational content to meet individual learning needs. Additionally, in customer service, memory-equipped agents can provide more efficient and personalized support, significantly enhancing user satisfaction and operational efficiency.

**Analogy to Human Brain** The architecture of memory in modern (M)LLMs is increasingly analogous to the synergistic relationship between different human brain systems, particularly the neocortex, the hippocampus, and the prefrontal cortex. This brain-inspired framework, which echoes the principles of Complementary Learning Systems theory (McClelland et al., 1995), provides a powerful lens through which to understand the different memory paradigms evolving in AI.

- Implicit Memory (§2): The Neocortex. We conceptualize the model's internal parameters as its digital neocortex. In the brain, the neocortex is the primary repository for long-term semantic knowledge, skills, and consolidated memories, which are learned slowly and stored in a distributed manner (Squire & Zola-Morgan, 1991). Similarly, a transformer's weights embody the implicit memory of the model—the foundational "world knowledge" acquired during pre-training. This parametric knowledge represents the model's stable, generalized understanding of language, patterns, and facts.

- Explicit Memory (§3): The Hippocampal System. To access specific, real-time, or episodic information, an AI system requires a mechanism analogous to the hippocampus. The hippocampus is critical for the rapid encoding of new episodic memories (*i.e.*, specific events and their context) and acts as an index that binds together disparate elements of an experience stored across the neocortex (Frankland & Bontempi, 2005). Explicit memory systems in AI, such as Retrieval-Augmented Generation (RAG), mimic this function. They serve as an "AI Hippocampus" by providing an on-demand, queryable index to external information (vector embeddings, knowledge graphs). This allows the model to ground its responses in specific, up-to-date facts without the need for slow, resource-intensive retraining of its entire parametric base (the "neocortex").

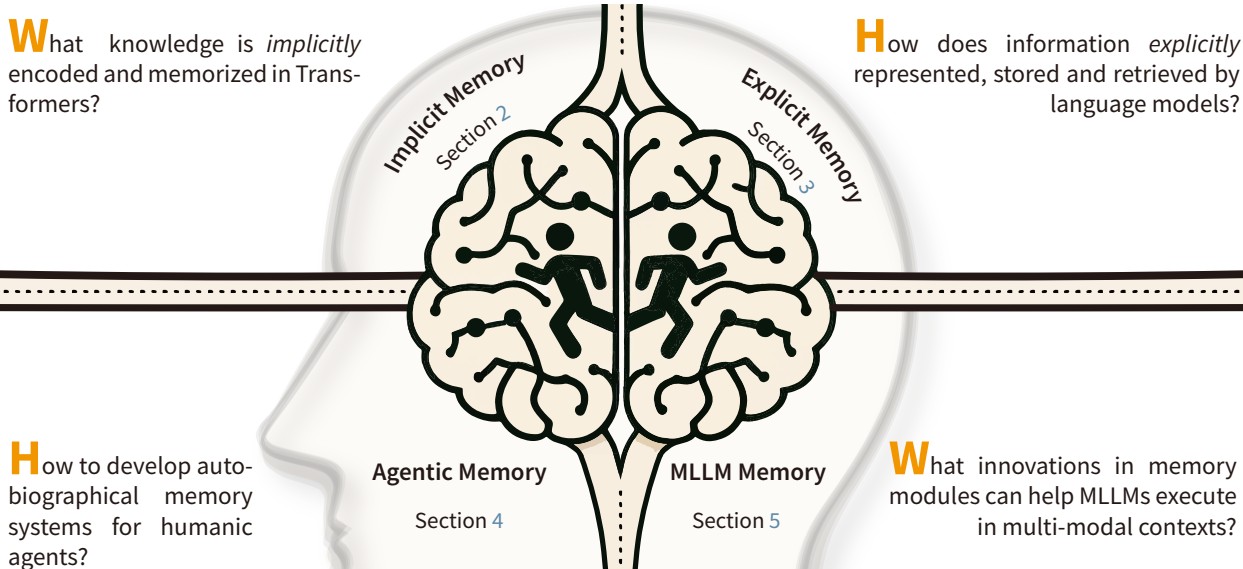

Figure 1: The overall framework of Memory Mechanisms in Large Language Models.

- Agentic Memory (§4): The Prefrontal Cortex. The functionality of agentic memory is best analogized to the prefrontal cortex (PFC), the brain's executive control center. The PFC is responsible for working memory, goal-directed planning, and integrating information from both long-term stores (neocortex) and recent episodic memories (hippocampus) to guide behavior (Miller & Cohen, 2001). Agentic memory systems similarly maintain a persistent state across interactions, manage working memory (*e.g.*, a scratchpad), and orchestrate the strategic retrieval and use of both implicit and explicit memory to formulate plans and execute complex tasks. Furthermore, as we explore in §5, this executive function extends to integrating information from specialized memory modules for spatial, temporal, and embodied intelligence, akin to how the PFC coordinates inputs from various sensory cortices.

This survey explores how these distinct yet interconnected memory paradigms are enhancing the journey toward more context-aware and intelligent AI systems.

**Related Surveys of Memory** Before the emergence of large language models (LLMs), Khosla et al. (2023) has explored memory-augmented neural networks. Their work investigated a range of network architectures, such as Hopfield Networks and Neural Turing Machines, and examined various types of memory, including sensory, short-term, and long-term memory. They also established connections between psychological theories of memory and their applications in AI, introducing architectures inspired by human memory systems.

Zhang et al. (2024d) presents a comprehensive survey on the memory mechanisms of LLM-based agents, systematically reviewing the design and evaluation of memory modules. Compared to our work, their focus is specifically on agents, particularly emphasizing historical and trajectory memory acquired through agent-environment interactions. He et al. (2024d) and Jiang et al. (2024a) focus on long-term memory in AI systems. Notably, Jiang et al. (2024a) proposes that AI equipped with long-term memory, capable of storing and managing real-world interactions, can achieve self-evolution.

More recently, increasing attention has been devoted to memory in AI systems. Du et al. (2025) reconceptualize memory systems by categorizing them according to atomic operations and representation types, distinguishing between parametric and contextual forms of memory. They further classify memory operations into management and utilization, offering a detailed taxonomy and technical analysis that provides practical insights. Shan et al. (2025) explores the similarities and differences between human memory and memory in LLMs, discussing various forms such as text-based, KV cache-based, parameter-based, and hidden-state-based

memory. Wu et al. (2025b) provides an in-depth analysis of memory in LLM-driven AI systems, categorizing memory-related methods across object, form, and temporal dimensions using an eight-quadrant framework.

While these surveys offer valuable perspectives, either drawing analogies with human memory or examining specific LLM-based memory forms and sources, none provide a unified view of memory study across pure LLMs, LLM-based agents, and further multimodal models. Consequently, our work presents a comprehensive survey that spans this full spectrum.

## 2 Implicit Memory: Unveiling Knowledge Inside Transformers

Implicit memory, a concept originating from psychology, refers to memories that are used unconsciously and are not stored explicitly Dew & Cabeza (2011). In the context of the deep Transformer era, we define "implicit memory" as follows:

> **Implicit Memory** refers to the intrinsic information embedded within a model's parameters, encompassing self-knowledge, facts, commonsense, associative memory, and other related elements, which collectively enable the generation of contextually relevant responses across a variety of tasks.

The rise of Transformer models (Kovaleva et al., 2019; Brown, 2020; Geva et al., 2022b; Zhang et al., 2022; Stolfo et al., 2024) has brought significant attention to implicit memory due to their remarkable performance across multiple domains. Recent research Li et al. (2025c) defines parameters of Transformer models as implicit long-term memory and the hidden states, KV-cache (in LLMs) as implicit short memory. Our survey focuses on exploring how these transformer models store and utilize knowledge within their parameters to understand and potentially enhance their capabilities. In this section, we attempt to answer the following research questions.

> **RQ1**: *What knowledge is implicitly encoded and memorized in Transformers?*
> **RQ2**: *How is information memorized, retrieved, and modified in Transformers?*

In the following subsections, we provide an overview of the current investigations on LLMs' memorization, including knowledge memorization and knowledge expression (§2.1.1), associative memory (§2.1.2), and implicit memory modification (§2.2). The structure of this section is demonstrated in Figure 2

### 2.1 Memory Analysis of Transformers

#### 2.1.1 Knowledge Memorization

Transformers (Vaswani, 2017) have shown an impressive ability to memorize and retrieve knowledge implicitly stored in their parameters (Geva et al., 2023; Yao et al., 2024b; Lv et al., 2024; Stolfo et al., 2024). Studies have investigated how components like Feed Forward Networks (FFNs) and Self-Attentions (SAs) contribute to this knowledge memorization (Kovaleva et al., 2019; Geva et al., 2021; 2022b; Dai et al., 2021; Clark, 2019; Hoover et al., 2020; Yu et al., 2023a). This research is crucial for comprehending the internal workings of Transformer models and, accordingly, to manipulate their stored knowledge to enhance model performance.

**Knowledge Memorized in Parameters** There are two primary hypotheses (**H1&H2**) concerning the memorization of knowledge in transformer-style language models.

> **H1:** Knowledge is encoded through FFNs, emphasizing the role of feed-forward layers within transformer architectures in memorizing information.

An FFN is usually implemented as a stack of interleaved linear and non-linear layers. It has been used as a sub-module in many different neural architectures and has been shown to be crucial for Transformer's representation power (Geva et al., 2021; Meng et al., 2022a; Gupta et al., 2023). We categorize memory mechanisms within FFNs into two classes:

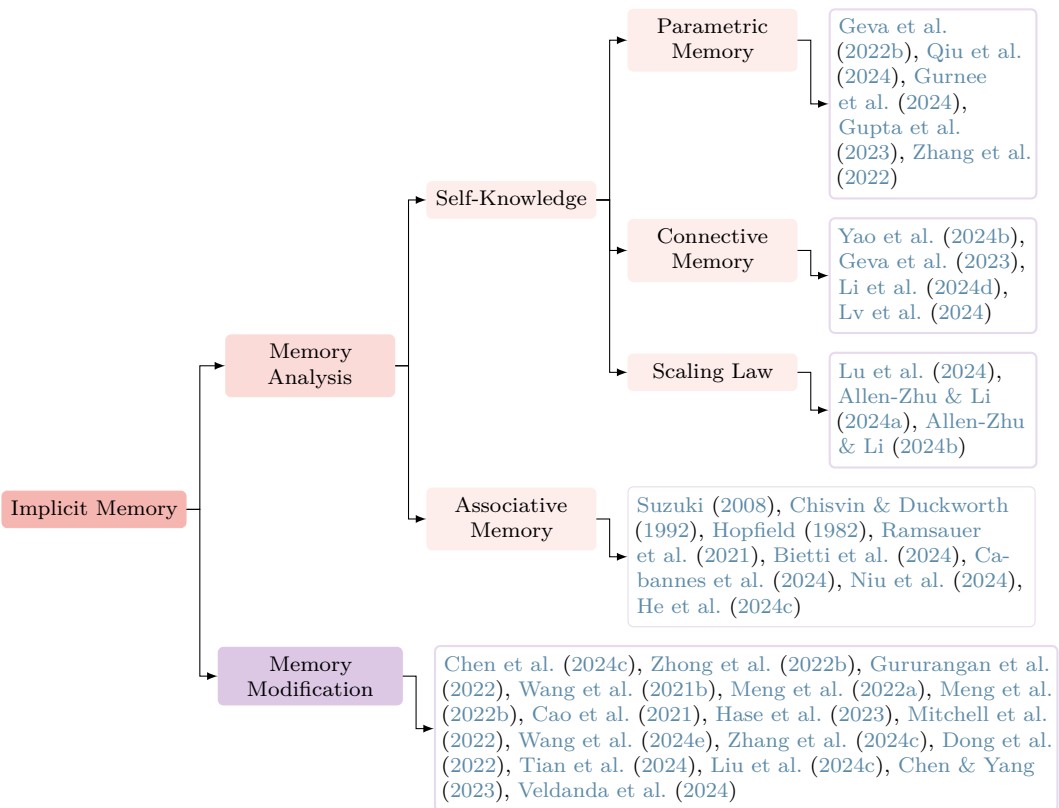

Figure 2: Taxonomy of implicit memory in Transformer.

*FFNs act as key-value memories.*[1] Geva et al. (2021) demonstrates that each key correlates with a specific set of human-interpretable textual patterns and each value induces a distribution over the output vocabulary. Based on this, Geva et al. (2022b) investigates the mechanism in which feed-forward layers update the inner representation, observing value vectors often encode human-interpretable concepts. Recently, Qiu et al. (2024) re-explores the key-value neural memories, conducting empirical ablation studies on updating keys or values in LLM. They recognize that updating the keys in a model is generally more effective than updating the values. Beyond these studies, Zhong et al. (2025) demonstrates, from a key-value memory perspective, why feed-forward networks (FFNs) adopt the ReLU kernel over more precise alternatives like the exponential kernel. They hypothesize that "A kernel with lower retrieval precision encourages a more polysemantic key–value memory: multiple unrelated facts can be stored under the same key space".

*Different FFN neurons memorize different information.* Dai et al. (2021) introduces the concept of knowledge neurons. They hypothesized that knowledge neurons in the FFN module are responsible for expressing facts. Zhang et al. (2022) shows that FFN neurons can be split into different functional partitions and some partitions are specialized in memorizing fact-related knowledge (Zhang et al., 2023f). Geva et al. (2022a) empirically finds that each vector of neurons in FFNs can be interpreted as a concept in the vocabulary space. Wu et al. (2023b) proposed a privacy neurons detector to locate neurons associated with private information. Stolfo et al. (2024) investigated entropy neurons and token frequency neurons which are two critical components believed to influence the representation and regulated uncertainty of LLMs.

**H2:** The attention mechanism is more crucial for knowledge storage, examining the relationship between the distribution of attention heads and the aggregation of knowledge.

---

[1]The feed-forward layer can be expressed as $\text{FF}(\mathbf{x}) = f(\mathbf{x} \cdot K^{\top}) \cdot V$, where $K$ denotes key vectors and $V$ denotes value vectors, distinct from the key-value pairs used in self-attention mechanisms.

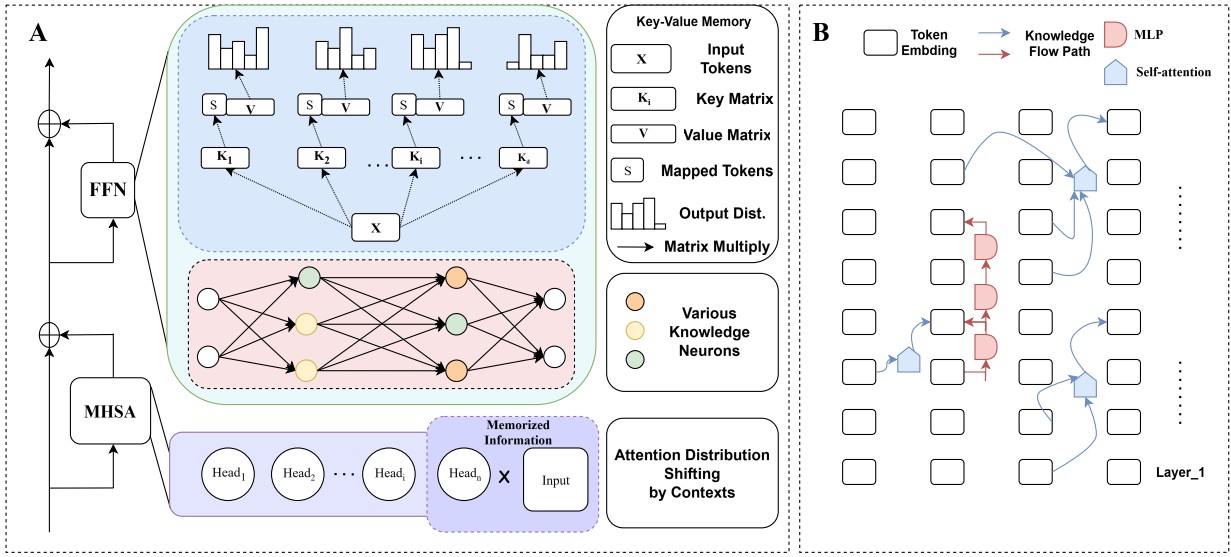

Figure 3: Parameter vs. Circuit. The left graph demonstrates the location of memory stored in Multi-Layer Perceptron layer and Self-Attention heads of the Transformer module, representing *FNNs act as key-value memories*, *Different FFN neurons memorize different information*, and *manipulating attention head distributions*, respectively. The right graph is a simplified demonstration of the knowledge flow between different transformer layers and various components within different layers.

Many works have analyzed Self-Attention layers for interpretability (Clark, 2019; Hoover et al., 2020), with *a focus on manipulating attention head distributions*. Recently, Yu et al. (2023a) study controlling LLMs to specifically leverage the in-context knowledge or facts memorized in pertaining by changing attention head distributions. Li et al. (2024c) identify a sparse set of attention heads that are completely related to fact knowledge of Alpaca Taori et al. (2023), and during inference, they shift activations along these truth-correlated directions, significantly improving the Alpaca truthfulness. Jiang et al. (2024c) further mathematically explores how Transformers can complete memory tasks based on the observation that LLMs' ability to retrieve facts can be easily manipulated by changing contexts. They theoretically prove and empirically show that the Transformer gathers information using self-attention.

**Knowledge Flows in Connections** As noted above, most studies have concentrated on knowledge storage within isolated components, such as FFNs and attention heads. In contrast, Yao et al. (2024b) studied connections between these Transformer components, introducing the concept of "knowledge circuits" to explore how different components collaborate to store and express knowledge. By ablating component-to-component connection, they demonstrate the knowledge circuit in LLMs for facts, linguistics, and commonsense. Previous work has explored knowledge flows similar to the knowledge circuit concept in LLMs. Geva et al. (2023) adopted the "knock out" strategy which blocks Multi-Layer Perceptron (MLP) or Multi-Head Self-Attention (MHSA) sublayers to investigate how the LLMs retrieved factual knowledge internally in inference. They conduct experiments on attribute prediction given subject-relation as queries, revealing two key components in the prediction process: the *subject enrichment process* where the early MLP sublayers are the primary source and the *attribute extraction operation* where the upper MHSA sublayers mainly carry out. Typically, Lv et al. (2024) explored several mechanisms employed by LLMs for factual recall tasks. They decompose MLP outputs into components that are easily understandable to humans based on linear regression, availably finding a universal anti-overconfidence mechanism in the final layer of models.

**Scaling Law of Knowledge Memorization.** The scaling law (Kaplan et al., 2020) has been used to describe model performance in terms of critical variables such as model size, dataset size, and the amount of computing used for training. *In the general pretraining field*, Kaplan et al. (2020) empirically study scaling laws for the performance of the language model on the cross-entropy loss $L$. They conclude an equation of

scaling laws with model size and training time:

$$L(N, S_{\min}) = \left(\frac{N_c}{N}\right)^{\alpha_N} + \left(\frac{S_c}{S_{\min}}\right)^{\alpha_S}, \tag{1}$$

where $N$ represents non-embedding parameters, $S_{\min}$ represents the minimum number of steps necessary to reach $L$, $\alpha_N \sim 0.077$, $\alpha_S \sim 0.76$, $N_c \sim 6.5 \times 10^{13}$, and $S_c \sim 2.1 \times 10^3$.

*For factual knowledge memorization*, Lu et al. (2024) investigates the relationship between model size, training epochs, and fact memorization. Their study reveals that LLMs' fact knowledge capacity follows a linear and negative exponential relationship with model size and training epochs, respectively, suggesting that memorizing all public facts, like those in Wikidata[2], is nearly impossible in a general pre-training setting:

$$C = C^* - \alpha_E \cdot \exp(-\beta_E \cdot Epoch), \tag{2}$$

where $C$ denotes fact capacity, $C^*$ means the LLMs' fact capacity saturation when epochs approach infinity, and $\alpha_E$ and $\beta_E$ are constants. Additionally, they find the scaling law of LLMs' fact memorization is similar to general pre-training, and the test loss $L$ on fact generalization also follows the power-law (Kaplan et al., 2020), promising the generalization of unseen fact knowledge:

$$L(D) = D_c * D^{\alpha_D}, \tag{3}$$

where $D$ is the number of training facts, $D_c$ and $\alpha_D$ are constant numbers. Their study also analyzes the compatibility and preference of LLMs' fact memorization, highlighting its inefficiency in handling redundant facts and its preference for memorizing more frequent or difficult facts. Allen-Zhu & Li (2024a) also explores the knowledge capacity scaling laws of LLMs, providing a more accurate and flexible alternative to traditional methods which often rely on evaluating language models against real-world benchmarks. Different from Lu et al. (2024), they use a synthetic dataset rather than real-world facts to avoid benchmark contamination. Through comparison across different model architectures and types of knowledge, they conclude that a fully trained Transformer can store 2 bits of knowledge per parameter, even when quantized to int8, which is close to the theoretical maximum. Further, Allen-Zhu & Li (2024b) observe that mixed training with raw knowledge text and question-answer pairs yields better performance on out-of-distribution questions compared to the pretraining-finetuning approach on their synthesized biography dataset. They conclude that rewriting the pretraining data for knowledge augmentation and integrating more instruction-finetuning data during the pretraining stage can enhance LLM's knowledge memorization and extraction.

### 2.1.2 Associative Memory

In psychology, associative memory is the ability to build relationships between two previously unrelated features or ideas such as phone number and the name of a person (Suzuki, 2008). There are also other researchers trying to use associative memory in the physical computer memory architecture to overcome some basic problems of traditional address-based memory (Chisvin & Duckworth, 1992). In Transformer-based models, associative memory refers to the memory of two previously unrelated representations (*e.g.*, input-output vectors) learned during the training process.

**Energy-based Model Mimic Associative Memory**  Hopfield Network (Hopfield, 1982) is a type of fullly-connected recurrent energy-based neural network that is used primarily for associative memory. This network structure leverages a set of interconnected neurons and weight matrices to encode multiple patterns, where each pattern is associated with a specific input. The storage of these patterns is facilitated by the modification of synaptic weights between neurons, a process that is governed by the principles of Hebbian learning. In details, the original Hopfield Network is composed with binary neurons $V_i \in \{0, 1\}$, so the instantaneous state of the system is defined by the combination of neurons' states $V_1, ..., V_n$. The initial value of the neuron states are all 0 as they are not "firing up" (Hopfield, 1982). The strength of one-way connection from neuron $V_j$ and neuron $V_i$ is denoted as $T_{ij}$ where $T_{ij}$ is computed using equation

$$T_{ij} = (2V_i - 1)(2V_j - 1) \; where \; T_{ii} = 0, \tag{4}$$

---

[2]https://www.wikidata.org/wiki/Wikidata:Main_Page

and the Non-connected neurons have strength $T_{ij} = 0$. The associative memory is stored in the format of patterns of states, and the connection strength $T_{ij}$ makes up the weight matrix $T$. The state of the system changes based on a pre-defined step-function at a given time $t$. The changes to a single neuron $V_i$ follows the step-function given the weight matrix elements $T_{ij}$ and all the neurons that connects to the neuron $V_i$

$$V_i = \begin{cases} 1 & \text{if } \sum_{j \neq i} T_{ij} V_j(t) > U_i \\ 0 & \text{if } \sum_{j \neq i} T_{ij} V_j(t) < U_i \end{cases} \tag{5}$$

where $U_i$ is a predefined threshold. Recent research on the Hopfield network (Ramsauer et al., 2021) has successfully integrated Dense Associative Memorys (DAMs) into modern deep learning architectures. This study introduces innovative updating rules and energy functions, transitioning the traditional discrete Hopfield network into a continuous framework. A key limitation of the original Hopfield network is its slow energy reduction during the pattern memorization phase. By utilizing a rectified polynomial energy function, DAMs accelerates energy decay, enabling the storage of more memory patterns within the same configuration space.

**Transformer-based Models**   Recent literature suggests that the Transformer architecture stores associative memories in the form of outer products of finite-dimensional embeddings within the intermediate weight matrices. Bietti et al. (2024) analyzes the transformer structures with an associative memory point of view. The authors construct a synthetic bi-gram dataset and a two-layer, single-attention-head Transformer model. Through empirical analysis, the results demonstrate that the Transformer architecture employs the weight matrix at the output of the attention block as a repository for associative memories. These associations enable the key-query matrices to direct attention to relevant tokens, thereby enhancing the model's inductive capabilities. The experiments also show how associations are learned through training dynamics and can be used to remap input to output vectors. Jiang et al. (2024b) explores how LLMs use tokens within a given context as memory clues to retrieve memory patterns from their parameters, and how context can be leveraged to influence or "hijack" the output of LLMs.

**Scaling Law of Associative Memory**   Cabannes et al. (2024) investigates the scaling law of the associative memory error rate in relation to model size and the number of data inputs, using a simple Transformer-based model. The error rate of the model is bounded by the previously seen data and the model capacity.

$$\mathcal{E}(f_q) \sim d^{-\alpha+1} + T^{-1+\frac{1}{\alpha}}, \tag{6}$$

where $d$ is the model capacity, and $T$ is the number of training samples. The model is denoted as $f_q$. $\alpha$ is the hyper-parameter that represent the Zipf exponent of the assumed data distribution. Compared to the exploration of scaling laws in knowledge memorization, Cabannes et al. (2024) focus on the embedding dimension $d$ as a measure of model capacity, whereas Kaplan et al. (2020) consider only non-embedding parameters. And in this formulation, the errors are quantified using the recall accuracy of previously presented factual inputs, while Kaplan et al. (2020) leverages cross-entropy loss. According to the equation, if the model possesses infinite memory, the error rate is constrained by the sample input size, as the model must learn sufficient associations to generalize the distribution. However, when memory is finite and the dataset size exceeds the memory capacity, the model attempts to remap new data into its memory space, which can result in memory interference when two distinct input-output relations are mapped to the same location. The paper also discusses various memory schemes for storing associations. Niu et al. 2024 conducted an empirical study on associative memory within the Transformer architecture and proposed a new energy function that, without introducing additional regularization terms, corresponds to a nearest-neighbor search over the memorized patterns.

**Usage of Associative Memory**   Numerous studies have investigated the use of associative memory as a mechanism for storing patterns linked to past data points. For instance, CAMELoT (He et al., 2024c) introduces an additional module within the original attention layer of a Transformer model to enhance its ability to handle longer context windows. The authors suggest that compressing past context or inputs into associative memory conserves memory by eliminating redundancy, thereby freeing up space for new content in fresh memory slots. This approach also facilitates the replacement of outdated memory slots with more recent inputs. Additionally, the authors show how modifications to the energy function and update rule enable the

modern Hopfield network to approximate the architecture of Transformer models. Krotov (2021) proposed an extension of the modern Hopfield network by incorporating an arbitrarily large number of recurrent layers, designed with a custom multi-layer recurrent model structure. Similarly, Millidge et al. (2022) proposed a generalized Hopfield network that decomposes a series of models—such as DAMs, Hopfield networks, and sparse distributed memories—into a three-stage framework comprising similarity, separation, and projection. This paper demonstrates that these three components not only generalize existing models but also extend their functional capabilities.

## 2.2 Implicit Memory Modification

Modifying implicit memory (Wang et al., 2024c) in language models involves altering the knowledge embedded within a model's parameters to enhance performance, reduce harmful outputs, and enable more efficient adaptation to new tasks. Focusing on methods for updating or removing knowledge without incurring the high costs of full retraining, as discussed in (Dai et al., 2021), we categorize memory modification into three main categories: **incremental training, memory editing, and memory unlearning**, as illustrated in Figure 4. **Incremental training** represents adding new knowledge into LLMs while building on the pre-existing information within them. **Memory editing** adjusts the embedded knowledge by modifying specific memory representations within LLMs. And **memory unlearning** eliminates incorrect or harmful internal knowledge from LLMs, thereby enhancing their reliability and trustworthiness. These approaches are intended to create more adaptable and efficient models, capable of rapidly incorporating new information while retaining consistent performance across various tasks. Techniques like dynamic updates, hyper-networks, and targeted unlearning play essential roles in enhancing the adaptability and dependability of LLMs for real-world applications.

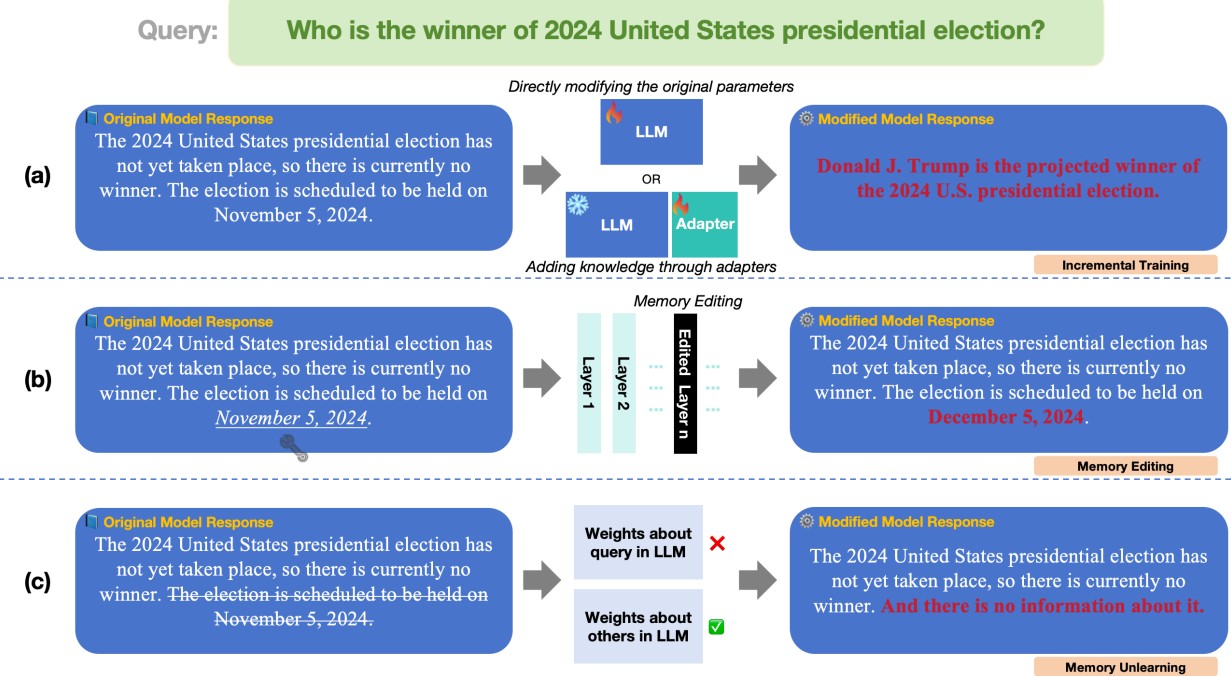

Figure 4: Three categories of Implicit Memory Modification

### 2.2.1 Modification Methods

**Incremental Training**    Incremental training in LLMs involves not only adding new knowledge but also ensuring consistency and alignment with the model's existing knowledge base. This process typically follows two main approaches: directly modifying the original parameters of the LLMs and adding knowledge through adapters, which store new memory separately.

*Directly modifying the original parameters.* Zhu et al. (2020) proposes a constrained fine-tuning method that selectively updates a subset of parameters to integrate new information efficiently. Expanding on Zhu et al. (2020), Padmanabhan et al. (2023) introduces context distillation to update knowledge using entity-specific texts while preserving the original LLMs distribution measured by KL-divergence. TRIME (Zhong et al., 2022b) incorporates in-batch examples as accessible memory during training. It improves performance by effectively leveraging local, long-term, and external memory with minimal computational overhead, achieving significant reductions in perplexity across various benchmarks. Another method, RECKONING (Chen et al., 2024c) encodes context-based knowledge into model parameters via a bi-level learning process, which consists of two loops: the inner loop adapts the model to memorize facts, while the outer loop uses the updated parameters to answer reasoning questions.

*Adding knowledge through adapters.* Adapters provide a flexible means of incorporating new knowledge by leaving the original model parameters untouched. LoRA (Hu et al., 2022) uses low-rank decomposition matrices for targeted updates while maintaining the model's core structure. K-ADAPTER (Wang et al., 2021b) adds separate adapters for different knowledge types in models like RoBERTa. DEMiX (Gururangan et al., 2022) introduces domain-specific expert networks, training only the relevant expert for new knowledge. These methods balance memory augmentation and model stability, making language models more adaptable and resource-efficient.

**Memory Editing**   The objective of knowledge editing is to incorporate new facts into a language model $\mathcal{M}_\theta$ through query-response pairs $\mathcal{D}_e = \{(q_i, x_i^*)\}_{i \in [1,N]}$. In this setup, $q$ is the query that triggers the retrieval of factual knowledge from $\mathcal{M}_\theta$, such as "The president of the US is", and $x$ is the intended response after editing, e.g., "Joe Biden". This integration is typically achieved by maximizing the probability of generating $x^*$ based on $q$, which can be expressed as:

$$\max_\theta p_\theta(x^*|q) \quad \text{where} \quad (q, x^*) \sim \mathcal{D}_e \tag{7}$$

Certain knowledge editing methods first identify the implicit memory within large language models (LLMs) that corresponds to the targeted knowledge, and then modify the knowledge stored in the model's weights. ROME (Meng et al., 2022a) and MEMIT (Meng et al., 2022b) use the causal trace method to identify the regions in LLMs where relevant knowledge about triplet relations is stored. These methods treat the middle-layer MLPs of GPT models as associative memory structures with key-value associations, and modify the weights of these layers to insert new associations. Dong et al. (2022) propose CKA, a method to detect incorrect knowledge in PLMs. CKA compares the model's scoring of correct facts against fake facts to determine if the model has accurately learned the facts. It also introduces a method called CALINET, which adds new parameters to the model while keeping the original parameters fixed, allowing it to learn the correct facts without overwriting other knowledge. KE (Cao et al., 2021) edits knowledge by employing a hyper-network to update the parameters of origin LLMs, so that LLMs can predict expected predictions for different inputs without affecting the prediction of any other input. Similarly, SLAG (Hase et al., 2023) utilizes a learnable hyper-network. This hyper-network takes model gradients as input and outputs new updates to be applied to the model parameters. Recently, the LAW (Wang et al., 2024e) method modifies specific MLP layer weights in a language model by adjusting the internal "key" and "values" associated with targeted knowledge, effectively disrupting the model's representation of that knowledge while preserving its reasoning abilities. FT-M (Zhang et al., 2024c) fine-tunes specific layers of the feed-forward network to maximize the probability of all tokens in the target sequence.

**Memory Unlearning**   Knowledge unlearning can be described as follows: Given a training set $D = \{(x, y)\}$ for the language model $\mathcal{M}_\theta$, where $x$ represents the input and $y$ represents the corresponding label, we define $D_f$ as the set of harmful and dangerous knowledge that we aim to forget, and $D_r$ as the set of data we wish to retain. The goal of knowledge unlearning is to enable $\mathcal{M}_\theta$ to remove all information from $D_f$ while preserving performance on $D_r$, which means:

$$\max_\theta dist\left(\mathcal{M}_\theta(D_f); \mathcal{M}'_\theta(D_f)\right) \quad \text{and} \quad \min_\theta dist\left(\mathcal{M}_\theta(D_r); \mathcal{M}'_\theta(D_r)\right) \tag{8}$$

Table 1: Comparison of Memory Editing Benchmarks Based on Different Dimensions.

| Dimension | KnowEdit(Zhang et al., 2024c) | MQuAKE(Zhong et al., 2023) | Eva-KELLM(Wu et al., 2023a) |
|---|---|---|---|
| knowledge insertion, modification, and erasure | ✓ | ✗ | ✗ |
| Counterfactual and temporal updates | ✗ | ✓ | ✓ |
| Supports multilingual data | ✗ | ✗ | ✓ |
| High-quality, validated datasets | ✓ | ✓ | ✓ |
| Multi-task editing capabilities | ✓ | ✗ | ✓ |
| Includes reasoning tasks | ✗ | ✓ | ✓ |
| Emphasizes cross-lingual performance | ✗ | ✗ | ✓ |
| Handles multi-hop reasoning | ✗ | ✓ | ✓ |
| Suitable for dynamic system updates | ✓ | ✓ | ✓ |
| Efficient evaluation time | ✓ | ✗ | ✓ |

Knowledge unlearning methods in LLMs can be categorized into distinct approaches based on their primary goals and techniques. Some methods, like a benchmark KnowUnDo (Tian et al., 2024) and a proposed method MemFlex (Tian et al., 2024) focus on precision unlearning, using targeted gradient manipulation to selectively remove sensitive information while preserving essential knowledge. In contrast, frameworks like SKU (Liu et al., 2024c) employ a two-stage approach, acquiring and systematically negating harmful knowledge to ensure safe responses while maintaining performance on benign tasks. Additionally, efficiency-focused methods, the Surgery framework (Veldanda et al., 2024) efficiently updates LLMs by unlearning outdated knowledge, integrating new information, and retaining performance on unchanged tasks using a three-part objective: reverse gradient for unlearning, gradient descent for updating, and KL divergence minimization for consistency. Lastly, solutions like EUL (Chen & Yang, 2023) prioritize selective and scalable unlearning by employing lightweight layers, facilitating iterative knowledge removal without impacting the model's overall capabilities, suitable for repeated applications where selective knowledge removal is essential.

### 2.2.2 Modification Benchmark

**Memory Editing Benchmark** Many memory editing benchmarks(Wang et al., 2023d; Cohen et al., 2024; Khandelwal et al., 2024) primarily focus on editing accuracy by constructing datasets designed to evaluate whether models can produce counterfactual responses when queried about specific factual knowledge. KnowEdit(Zhang et al., 2024c), for instance, includes a suite of six datasets tailored for assessing various knowledge editing methods. These datasets cover a diverse range of editing types, such as fact manipulation, sentiment alteration, and hallucination generation. This benchmark consolidates key evaluation criteria into four categories: edit success, portability, locality, and fluency, thereby providing a comprehensive evaluation framework for different editing approaches. MQuAKE(Zhong et al., 2023) offers a distinct perspective by evaluating whether edited models can answer multi-hop questions where the answer should logically change as an entailed consequence, revealing the limitations of prior methods for such questions. Eva-KELLM(Wu et al., 2023a) expands the scope of knowledge editing evaluation to a more general scenario where raw documents within datasets are directly utilized for editing. This benchmark offers greater generality and practical relevance, allowing for the assessment of various knowledge editing methods' performance in a multilingual context. The comparison of these three benchmarks can be seen in table 1.

### 2.3 Limitations, Open Questions, Discussion

Current research in implicit memory often faces several key challenges:

- Generalization of findings: A significant limitation of many studies is that they focus primarily on knowledge memorization and extraction within the confines of specific tasks (e.g., relation triples prediction) or particular types of knowledge (such as facts, commonsense, or bias-related knowledge). This narrow focus does not establish the generalizability or broader applicability of the conclusions drawn.

- Efficiency of probing methods: Research on knowledge circuit exploration is often hindered by the time-consuming nature of conducting numerous component-to-component ablations. This complexity can significantly slow down the process of systematically investigating model behavior and knowledge extractions.

- Risk of knowledge unlearning: While knowledge unlearning aims to remove unwanted or harmful information, it carries the potential risk of inadvertently disrupting related knowledge. This disruption can lead to unintended consequences and degraded performance in downstream tasks. Consequently, more comprehensive evaluations of models subjected to knowledge unlearning are needed to fully understand these effects.

Looking ahead, future research should focus on gaining a deeper understanding of the internal mechanisms of Transformer models and exploring the development of more effective and efficient computational frameworks for implicit memory modeling.

## 3 Explicit Memory: When (M)LLMs Meet Retrieval

> In this work, we refer to **explicit memory** as specific, structured, or unstructured representations that store factual knowledge, history trajectories in an external storage.

Explicit memory allows the retriever to dynamically capture context-aware information and enables the generator to adaptively incorporate knowledge from external memory, thus improving the quality of generated outputs. It facilitates the retention and retrieval of information across sessions, ensuring continuity and enhancing the model's capacity to handle long contexts without exceeding its input limitations. Explicit memory plays a crucial role in providing flexibility and enhancing interpretability, especially in interactive, knowledge-intensive, and rapidly evolving domains.

In this section, we focus on the research question:

*How is the explicit memory represented and utilized in different training and application scenarios?*

We will discuss how different types of memory are **externally represented and stored** (§3.1) the **interaction and updating of external knowledge during training** (§3.2), and how to **externalize implicit knowledge for retrieval** in typical scenarios (§3.3).

### 3.1 Explicit Memory Representation

### 3.1.1 Free text

LLMs are usually trained on free-form text, depending on target tasks. The free-form text is represented and stored at different levels of granularity as shown in Fig. 6:

**Document**   As the largest memory unit, a document retains rich contextual information, which is crucial for understanding the overarching themes (Ren et al., 2023a). Commonly employed retrieval methods can be broadly categorized into sparse retriever and dense retriever. Sparse retrieval, exemplified by traditional search engines and the BM25 algorithm, relies on exact matches and TF-IDF weighting to retrieve lexically relevant documents (Asai et al., 2024; Ma et al., 2023). In contrast, dense retrieval focuses on semantic similarity (Shi et al., 2024; Melz, 2023) and is typically implemented using tools such as FAISS (Johnson et al., 2021) and pre-trained encoders. Once retrieved, this supplementary knowledge is concatenated with the

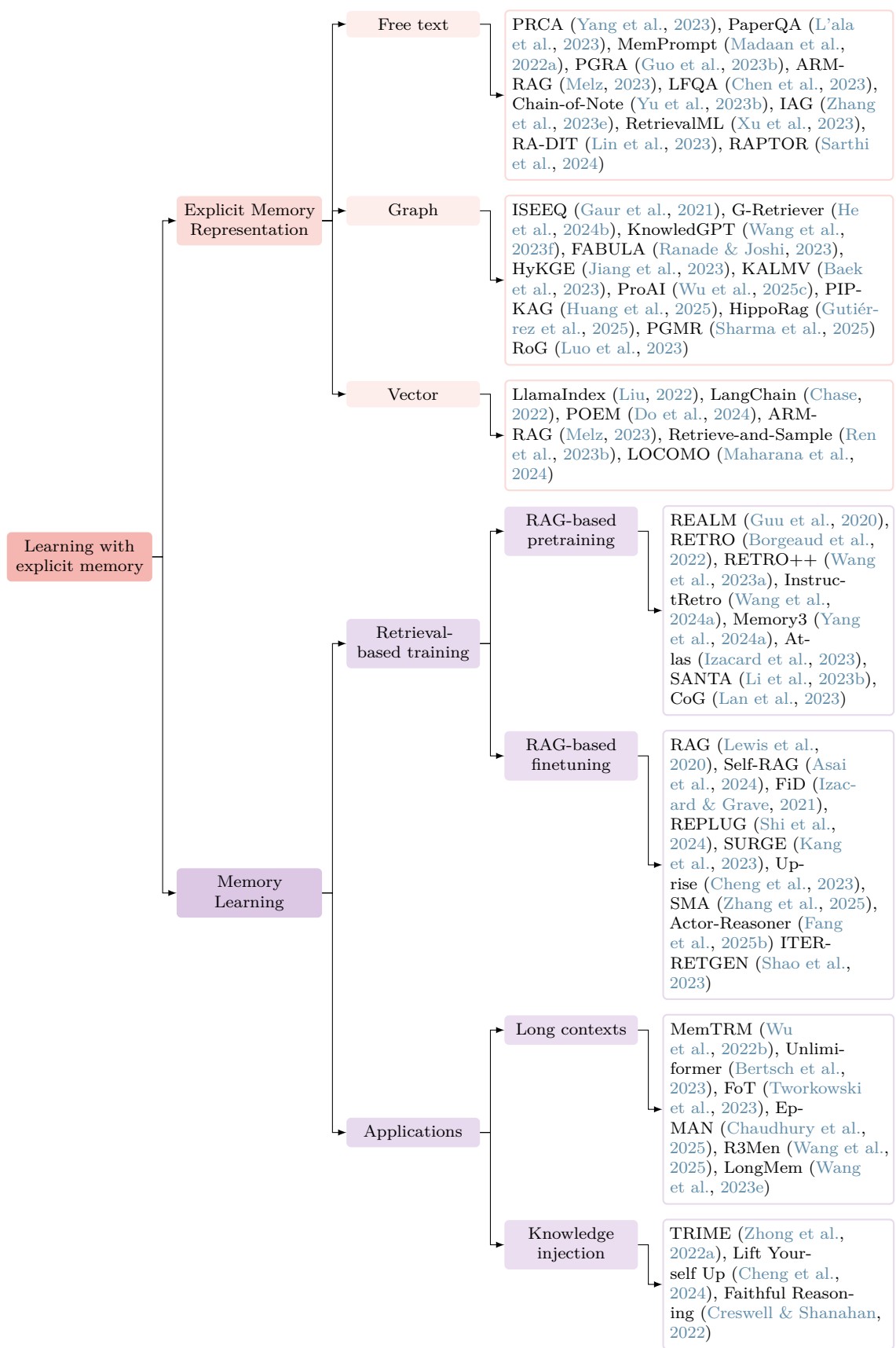

Figure 5: Taxonomy of the structure design for learning with explicit memory.

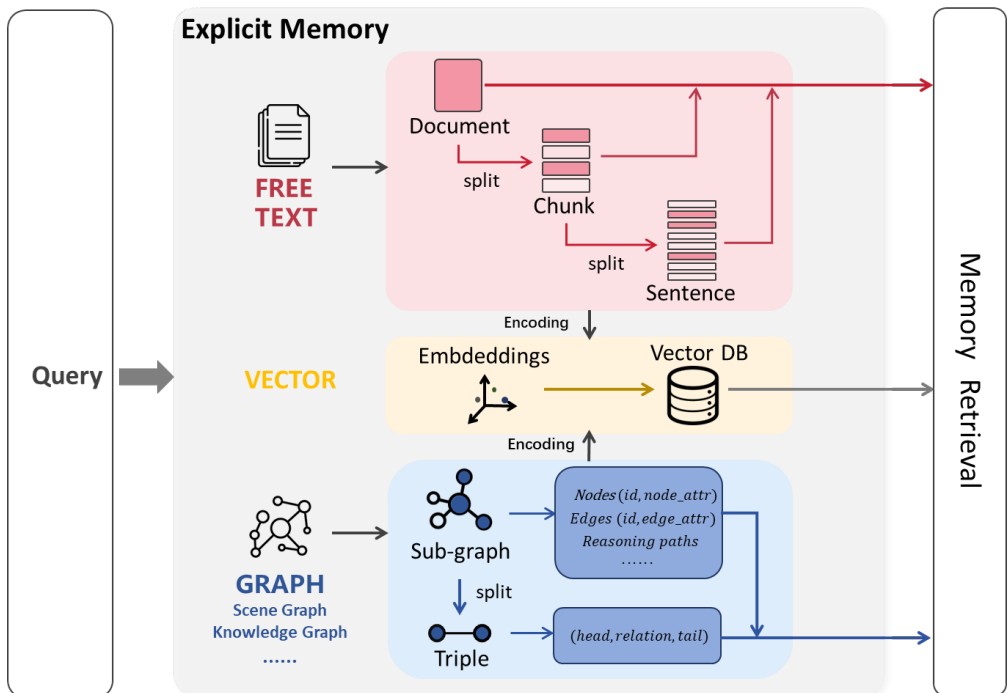

Figure 6: Three types of representations for explicit memory retrieval

question as input to the model. However, overly-long texts can introduce noise (Yang et al., 2023; Chen et al., 2023; Cuconasu et al., 2024; Zhang et al., 2023e), potentially obscuring specific facts (Trivedi et al., 2022; Yu et al., 2023b). Moreover, using entire documents as input increases inference time and computational costs for LLMs. Due to limited context window size in LLMs, excessively long texts will be truncated, risking the omission of critical data.

**Chunk** Dividing text into fixed-size chunks effectively reduces redundancy and strengthens the connection between retrieved memory and the query (Wang et al., 2024a; 2023a; Lin et al., 2023). There can be multiple chunks relevant to the query, but using all of them is usually infeasible due to the context length limit. To address this, RetrievalML (Xu et al., 2023) selects the top k chunks based on relevance, determining the optimal k value through experimentation. In contrast, PaperQA (L'ala et al., 2023) employs LLM-generated relevance scores between the chunks and the query for more refined filtering. To preserve textual coherence, RAPTOR (Sarthi et al., 2024) clusters the chunks and generates summaries for each cluster. While chunking may disrupt semantic continuity and omit details within chunks, it offers a practical balance between preserving semantic integrity and enabling LLMs to comprehend specific details.

**Sentence** A sentence-level memory unit (Cheng et al., 2024) enables the capture of specific facts (Wang et al., 2023i) and details but tends to overlook connections between sentences, potentially leading to discontinuities in understanding. Thus, sentence-level segmentation is typically employed in scenarios requiring a detailed grasp of individual sentences' meanings, such as sentiment analysis (Guo et al., 2023b) and knowledge editing (Zhong et al.). In addition, MemPrompt (Madaan et al., 2022a) uses a sentence to assist the model in understanding specific tasks.

### 3.1.2 Graph

Graph-based data organization employs nodes and edges to structure knowledge systematically. Nodes represent distinct units of information, which can range from words and sentences to entire paragraphs. Edges, on the other hand, signify the relationships between these units, illustrating how they are interconnected. This structured approach to data organization proves particularly advantageous for tasks requiring advanced

reasoning capabilities. For instance, reasoning through graphs enables flexible transitions between different reasoning paths, accommodating both logical deduction and multi-hop reasoning. Multi-hop reasoning involves traversing multiple interconnected nodes to infer complex relationships or derive conclusions that span diverse pieces of information. By leveraging this graph-based structure, one can achieve more efficient and nuanced reasoning processes. In a recent development, HippoRAG (Gutiérrez et al., 2025) incorporates the Personalized PageRank algorithm along with the inherent ability of an LLM to automatically build a knowledge graph, thereby enhancing the retrieval process with multi-hop reasoning capabilities.

**Sub-graph**  Sub-graphs represent specific portions of the overall graph that are most closely related to the query (Ranade & Joshi, 2023). The nodes in a sub-graph may represent entities, sentences, or paragraphs. G-Retriever (He et al., 2024b) takes a textual graph describing nodes and edges as part of the input to use the structured information in the graph. HyKGE (Jiang et al., 2023) searches for sub-graphs based on the entities mentioned in the query and extracts the reasoning path to improve LLM's reasoning ability. The primary challenge of using sub-graphs lies in the high costs of constructing and maintaining sub-graphs.

**Triple**  Triples are the fundamental units of graphs, representing entities and their relationships in a structured format (Wang et al., 2023f; Kang et al., 2023). They provide fine-grained knowledge but are limited by their fragmented nature, i.e., a triple only involves two out of a large number of entities, which may hinder the expression of continuous semantic information unless enough triples are retrieved. To use this precise memory, ISEEQ (Gaur et al., 2021)inserts triple facts into the appropriate position of the original query. KnowledGPT (Wang et al., 2023f) goes a step further and adds additional descriptions of the entities. Because this type of memory is concise and explicit, the retrieved irrelevant triples will directly affect the correctness of the generated content. Therefore, KALMV (Baek et al., 2023) proposes a detection method for this problem. Besides, PGMR (Sharma et al., 2025), a new modular architecture featuring a non-parametric memory retriever module for managing KG elements, thereby enhancing the accuracy and reliability of SPARQL queries generated by LLMs.

### 3.1.3 Vector

Vectors (Liu, 2022; Cheng et al., 2024; Chase, 2022; Yang et al., 2024a; Borgeaud et al., 2022; Do et al., 2024) characterize rich semantics and thus facilitate improved contextual understanding of LLMs.

Original text is segmented into smaller fragments that are then converted into vector representations through an encoding model and archived in a vector-based knowledge base. Typically given a query, recent works (Melz, 2023; Ren et al., 2023b; Maharana et al., 2024; Lin et al., 2023) retrieve relevant knowledge from the memory by computing the similarity between the query and the stored embeddings. The top K fragments with the highest similarity will be used as part of the prompt to improve the LLMs' understanding of the query.

As a memory format, vectors offer three **advantages** over free text and graph-based representations:

- **Robust semantic understanding** Vectors represent words, sentences, or even graph nodes in a continuous space where semantically similar entities are closer to each other. However, semantic relationships among free texts or entities in the graph are not apparently captured and need to be further inferred.
- **Scalability and Flexibility** Vectors can handle large-scale data with more efficient indexing, quantization, and batch processing, circumventing the limitations of simple keyword matching. The other two representations requires heavy pre-processing and more storage space suffering from higher complexity.
- **Generalization and Transfer Learning** Vectors capture underlying semantic similarity and can generalize to different tasks and domains with relatively little retraining. In addition, it is capable of processing multi-modal data, such as images and audio. Free-form text is typically task-specific, requiring manually crafted features, while graph structures are usually associated with specific topologies and require corresponding graphical algorithms.

### 3.2 Training with Explicit Memory

Training on explicit memory enables the model to efficiently utilize external memory by retrieving, adapting, and refining it, avoiding the need to reprocess the entire dataset. It involves learning how to interact, organize,

and integrate well-structured memory representations during training rather than relying solely on retrieval based on memory similarity or structural matching, which can lead to redundancy.

There are several key **advantages** compared to training-free retrieval. Firstly, it optimizes retrieval relevance and generation accuracy by training each module for more context-sensitive use and alignment. Additionally, the training process enables LLMs to perform more sophisticated inferences by integrating external past experiences with the model's inherent reasoning capability. Specifically, for scenarios with long context that require consistency or accuracy across multiple turns of interactions, it ensures consistent responses over successive queries and adapt to rapidly evolving information or user-specific scenarios.

In this subsection, we distinguish two critical training phases: **pre-training** (§3.2.1) and **fine-tuning** (§3.2.2). We aim to explain how both stages contribute to the enhanced performance of models equipped with explicit memory systems. Besides, we introduce a few works to address the following three **main challenges**: i) **high computational costs** of real-time encoding of large-scale retrieved texts; ii) **lack of interpretability** in retrieval during training and inference; iii) **poor performance** in retrieval and generation when training autonomously. Tree typical pipelines for training with explicit memory can be seen in Fig. 7.

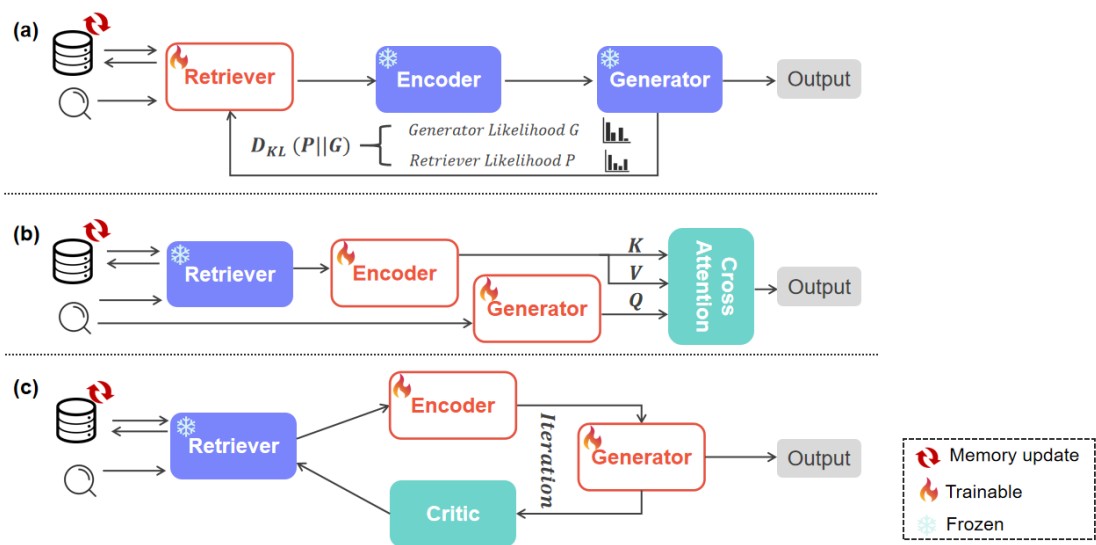

Figure 7: Three typical pipelines for training with explicit memory.

### 3.2.1 Pre-Training

Pre-training involves training an LLM from amounts of diverse text data by predicting the next token. The goal is to obtain an LLM with a comprehensive understanding of human languages and a general task-solving ability, thereby establishing a robust foundation for subsequent knowledge-intensive tasks and enhancing their retrieval capabilities.

**Unsupervised Memory Retrieval** REALM (Guu et al., 2020) proposed an approach to knowledge storage, successfully pre-training a knowledge retriever in an unsupervised manner for the first time. Specifically, it achieves end-to-end training by modeling both the retrieval and prediction processes. This work demonstrated significant performance improvements in open-domain question-answering tasks, introducing the first unsupervised pre-training method for a knowledge retriever using masked language modeling and backpropagation.

**Advances in Retrieval Integration**  RETRO (Borgeaud et al., 2022), unlike REALM, directly appends retrieved content to the prompt and integrates retrieved data via chunked cross-attention. RETRO employs a pre-trained BERT as the retriever and freezes it during pre-training. This approach significantly enhances the memory capacity of LLM without increasing computational overhead, enabling the efficient integration of external knowledge at a much larger scale. Subsequently, RETRO++ (Wang et al., 2023a) analyzes RETRO and GPT models, illustrating the advantages of retrieval-augmented architectures in text generation and zero-shot knowledge-intensive tasks. Building on these insights, InstructRetro (Wang et al., 2024a) applied the RETRO framework to pre-training and instruction-tuning of GPT, resulting in improved accuracy and generalizability on complex, knowledge-intensive tasks.

**Innovations in Knowledge-Intensive Pre-Training**  Memory[3] (Yang et al., 2024a) leverages a two-stage pre-training strategy by selectively storing key-value pairs with lower read and write costs. Memory[3] demonstrated its potential to enhance both the efficiency and performance of LLM across various tasks. Additionally, various pre-training methodologies have been proposed to address other challenges in LLM development. For example, Atlas (Izacard et al., 2023) employed a dual-encoder architecture for dense retrieval, combined with a sequence-to-sequence model, then used joint pre-training to better integrate retrieved documents for knowledge-intensive tasks. SANTA (Li et al., 2023b) utilized structured data alignment and masked entity prediction as pre-training techniques, achieving state-of-the-art performance in code and product search tasks. To maintain coherence and diversity in generation, CoG (Lan et al., 2023) generates text progressively copying fragments from existing corpora, outperforming traditional and retrieval-augmented models across multiple evaluations.

### 3.2.2 Fine-Tuning

Fine-tuning on the explicit memory allows the model to be specialized with task-specific or domain-specific knowledge, ensuring that it can handle nuanced information retrieval in targeted applications and scenarios.

A notable work is the Retrieval-Augmented Generation (RAG) model proposed by Lewis et al. (2020), which combines parametric memory from a pre-trained seq2seq model with non-parametric memory, such as a dense vector index of Wikipedia. By jointly finetuning the retriever and generator, RAG effectively learns to retrieve relevant information and condition its output on external knowledge, significantly improving factuality, diversity, and specificity in generated responses. This finetuning strategy has led to substantial performance improvements across knowledge-intensive tasks, such as open-domain question answering and fact verification.

**Autonomous Memory Retrieval**  Building on RAG, Self-RAG (Asai et al., 2024) introduced a self-reflection mechanism that further enhances retrieval and generation processes. During training, a critic model generates reflection tokens, which are inserted into the training data. These tokens help the generator learn when and how to retrieve relevant information more intelligently. By finetuning this self-reflective system, Self-RAG shows significant advantages in tasks that require factual verification, reasoning, and long-text generation, allowing the model to be more efficient and accurate in deciding when retrieval is necessary and how to use the retrieved content effectively. Besides, SMA (Zhang et al., 2025)introduces self-memory alignment to enhance the generalization of LLMs and balance trade-offs between different capabilities. Specifically, it fine-tunes the model on self-generated responses to precise, simple factual questions using preference optimization. Extensive experiments demonstrate that SMA significantly improves the overall performance of LLMs, consistently enhancing factual accuracy, helpfulness, and comprehensive skills across various benchmarks.

**Context-Aware and Task-Driven Memory Retrieval**  To further improve task-specific performance, UPRISE (Cheng et al., 2023) finetunes a lightweight prompt retriever to automatically retrieve prompts tailored to specific inputs, improving LLMs' zero-shot capabilities of long-form question-answering. For context-aware dialogue generation, SURGE (Kang et al., 2023) leverages subgraph retrieval through graph-text contrastive learning. By finetuning both the subgraph retriever and the generator, SURGE enhances the model's ability to handle complex, structured data within dialogues.

**Efficient Large-Scale Memory Retrieval**   As processing large amounts of retrieved text passages can increase computational costs, methods such as Fusion-in-Decoder (FiD) (Izacard & Grave, 2021) have been proposed to address these challenges. FiD processes each retrieved passage independently in the encoder while jointly processing them in the decoder, thus optimizing computational efficiency. Similarly, REPLUG (Shi et al., 2024) introduces a retrieval-augmented approach that integrates relevant documents with input context without modifying the internal parameters of the LLM. This reduces the computational burden of finetuning large models (e.g., 405B parameters) by minimizing the KL divergence between retrieval likelihood and the model's output perplexity.

## 3.3   Training with externalized parameteric knowledge

> **Externalized Parametric Knowledge** effectively extracts and externally stores portions of the model's internal knowledge or its own intermediate outputs in a structured and accessible format.

It is particularly useful in tasks that involve processing **long documents**(§3.3.1) and **Knowledge injection**(§3.3.2), where the model's internal memory is insufficient for handling the entire context. It can retrieve the information as required, allowing for the efficient handling of extended contexts while maintaining coherence and accuracy throughout the task. This mechanism serves as a bridge between the model's ability to handle inherent knowledge and its ability to maintain and process large volumes of external information over time, avoiding the limitations of using internal memory alone. This method provides additional flexibility in accessing and utilizing information, ensuring that when the model's internal memory is not sufficient for a given task, explicit external storage comes to its rescue.

### 3.3.1   Long Contexts

The traditional Transformer architecture faces significant challenges in capturing long-range dependencies due to the limited context length imposed by the attention mechanism (Li et al., 2023a; Wu et al., 2025a). Yet, many tasks require models to process distant information, which is often crucial for accurate predictions. To address this limitation, Wu et al. (2022b) introduced MemTRM, a language model designed to memorize representations of previous inputs. MemTRM stores key-value pairs of past inputs and uses approximate k-nearest neighbor (kNN) search to extend the model's effective attention span. It demonstrates that MemTRM significantly improves performance across various tasks by expanding the model's attention context.

Building on this concept, several works take different approaches to extending the attention span. One typical framework leveraging external memory for long context can be seen in Fig. 8. One such model is Unlimiformer (Bertsch et al., 2023), which offloads cross-attention computation to a kNN index. Unlike MemTRM, Unlimiformer is fully non-parametric, requiring no fine-tuning, and allows each attention head in every decoder layer to focus only on its top-k keys. This attention reconstruction capability enables Unlimiformer to perform personalized retrieval in each layer while maintaining greater efficiency than MemTRM. Another related work is FoT (Tworkowski et al., 2023), which extends the model's context length through fine-tuning rather than modifying the architecture. FoT has shown significant promise in enhancing the ability of LLM to handle long-text tasks effectively. A recent work EpMAN (Chaudhury et al., 2025) proposes an architecture combining episodic memory attention with self-attention during LLM training for robust long context performance.

Despite the advancements of MemTRM and its derivatives, the coupled memory design in MemTRM presents a challenge: the cached representations of past inputs may diverge from the current model representations as model parameters are updated. This distribution shift limits the effectiveness of memory-augmented models over time. To address this issue, LongMem (Wang et al., 2023e) decouples the network architecture by freezing the original LLM as a memory encoder and introducing an adaptive residual side network to act as the memory retriever and reader. This decoupling not only mitigates the issues caused by distribution shifts but also demonstrates superior performance in long-text processing and contextual learning tasks.

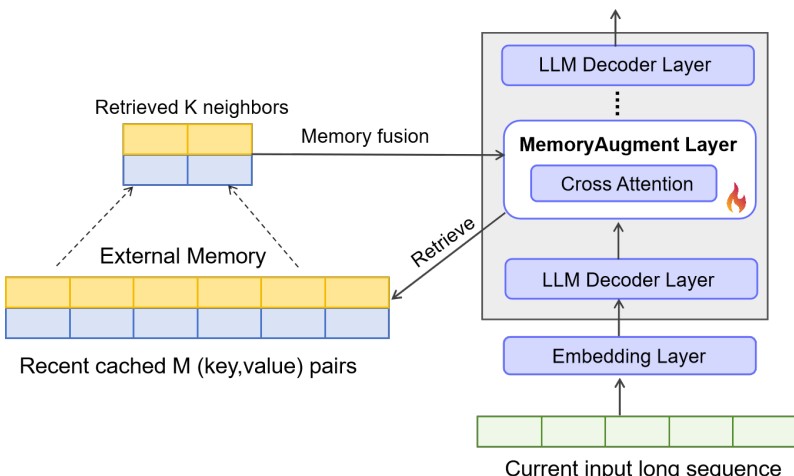

Figure 8: External memory training for long context.

### 3.3.2 Knowledge Injection

There are other research works leverages knowledge provided as part of the context from memory retrieval for knowledge injection and augmentation. These works aim to reason more robustly by folding the provided contextual knowledge into the model's parameters and complete the downstream tasks by using the updated parameters.

TRIME (Zhong et al., 2022a) utilizes a contrastive learning objective that aligns the hidden representation of a token with both token embeddings and a set of in-batch contextualized representations. The method introduces new strategies for memory construction and data batching to adapt to different memory types at testing time, which allows for back propagation to all memory representations. Similarly with various memory types, Yogatama et al. (2021) introduces an adaptive semi-parametric language model (SPALM) that integrates a non-parametric episodic memory component with extended short-term context through cached local hidden states and global long-term memory by retrieving nearest neighbor tokens at each timestep. A gating function is designed to adaptively combine information from local context, short-term memory, and long-term memory.

There are other works encoding text of different levels of granularity into embedding for retrieval. TOME (De Jong et al., 2021) integrates a semi-parametric representation into its architecture as a source of factual knowledge through "mention memory". It maintains a table of dense vector representations for every entity mentioned in a corpus, allowing TOME to retrieve and assimilate information from multiple sources making it more scalable and efficient compared. Open Predicate Query Language (OPQL) (Sun et al., 2021) uses a dual-encoder pre-training process to encode relation mentions, which can be integrated into a language model (OPQL-LM) to improve performance on open-domain question answering. Unlike previous methods that rely on distant supervision from a knowledge base, OPQL is a method for constructing a virtual knowledge base from text without any structured supervision.

### 3.4 Limitations, Open Questions, Discussion

To continuously learn, adapt, and improve the ability of LLMs to learn with explicit memory, we identify the following open questions and encourage future research to address them.

**Can RAG solve the limitation of long context?** Enhancing the long context capabilities of LLMs is crucial in downstream tasks. There are currently two main approaches when handling longer sequences: extending an LMs context length or RAG. Training LLMs with extended context windows allows them to process long inputs continuously without truncation, thus improving their attention span. However, this also incurs higher computational costs and memory requirements. Methods using RAG dynamically retrieve concise, relevant documents based on the query, without the need to store the entire input. Recent studies have positioned LLMs as a promising paradigm for time-series analytics and spatio-temporal data science

Liu et al. (2025a;b); Liang et al. (2025b). Both domains are characterized by extremely long historical sequences, which can impose substantial computational and storage burdens. The integration of explicit memory mechanisms offers a potential solution by mitigating the need to store the entire historical context, thereby enhancing efficiency in these applications. Further research is needed to develop mechanisms beyond simple semantic matching, which can create a more robust and dynamic memory system while avoiding memory overload.

A deeper analysis of potential applications for each method, and how they can be integrated into a unified and flexible system, offers a promising direction for future research.

**When and how to retrieve more intelligently and autonomously** The development of more intelligent and autonomous memory retrieval mechanisms is crucial for optimizing tasks such as retrieval, reasoning and content generation. This enables the model to efficiently decide when retrieval is necessary and how to integrate the retrieved memory seamlessly for improved performance. An intelligent retrieval process is crucial for adaptively absorbing contextual information from external memory based on task-specific needs, especially for tasks requiring complex interactions with various data sources.

**How to enhance retrieval-based training to avoid hallucination and contamination** Incorporating explicit memory for knowledge injection during training allows LLMs to access relevant information more precisely and adaptively, rather than relying solely on parametric knowledge or context provided at generation time. However, LLMs may encounter memory contamination, where irrelevant or incorrect information is unintentionally stored during the learning process. A promising direction for future research is the design of selective memory mechanisms that filter out irrelevant details while retaining crucial information, thus reducing model hallucinations and preventing memory contamination.

In addition to these broad research topics, there are several engineering challenges that need to be addressed. For example, developing a retrieval method that maintains **consistency and coherence between external memory and the implicit memory** learned previously, without disruption over time. Incorporating large-scale retrieval-augmented models during training presents a significant challenge. The retriever must consider millions of candidate documents and backpropagate, which requires substantial computational resources. This presents an important direction for future work: optimizing the model structure, storage, and document selection to reduce computational burden and enhance **scalability and efficiency**.

## 4 Agentic Memory: Consolidating Memories into Humanic Agents

In the study of cognitive science, memory encompasses the cognitive processes involved in encoding, storing, and retrieving information. According to the Atkinson-Shiffrin three-stage memory model (Atkinson, 1968), information in the human brain progresses through three distinct stages: it initially enters sensory memory, then transitions to short-term memory, and ultimately consolidates into long-term memory, as illustrated in Figure 9.

LLM agents are artificial intelligence systems that understand and generate human-like text within real-world environments. These agents are capable of performing tasks such as answering questions and engaging in conversations. One of the central functionalities of LLM agents is their memory system, which mirrors the structure and processes of human memory. With memory modules, LLM agents can accumulate experiences, adapt continuously, and exhibit consistent, rational, and effective behaviors. Specifically, the memory modules in LLM agents facilitate the retention of past interactions and knowledge, enabling the agents to reference previous information and improve their performance over time. In this context, we categorize the memory systems of LLM agents in a manner analogous to human memory, as follows:

> **Sensory Memory** is the initial stage of memory, responsible for briefly retaining sensory information. It includes iconic (visual) memory, echoic (auditory) memory, and haptic (touch) memory. In the context of LLM agents, we regard Sensory Memory as the data ingestion pipeline of AI systems, such as DataLoader, and will not discuss it in detail.

> **Short-term Memory (STM)** temporarily stores the information currently in awareness, as well as information necessary for complex cognitive tasks such as learning and reasoning[a]. For LLM agents, STM refers to the information maintained in the context window during in-context learning, which is thus constrained by the limited context window length of the Transformer architecture.
>
> ───────────────
> [a]Typically stores about 7 items, with a duration of approximately 20-30 seconds.

> **Long-term Memory (LTM)** stores information for long periods in human cognition[a]. One type of LTM is declarative memory of facts and events, which can be consciously recalled. Another type is procedural memory including unconscious skills. For LLM agents, LTM serves as an external storage system, accessible to the agent during queries through efficient retrieval mechanisms.
>
> ───────────────
> [a]Typically with virtually unlimited capacity, lasting from days to decades

Although Zhang et al. (2024d) has examined the reasons, contents, and methods of storing memories in LLM agents, a comprehensive categorization of memory usage approaches based on an analogy to human cognitive processes, along with a review of engineering-level memory systems for agents available on the market, remains absent.

In this section, we will examine: (1) how an LLM agent's memory system operates across both **short-term and long-term** contexts (§ 4.1), drawing inspiration from human cognitive processes; (2) the mechanisms through which **multiple agents** share memories (§ 4.2); (3) the pipeline for **data ingestion, storing, indexing, and application** within agent memory systems (§ 4.3); and (4) methodologies for **evaluating** the effectiveness of these memory mechanisms (§ 4.4).

## 4.1 Single-agent Memory

Recently, several studies have been conducted to augment LLM agents with non-parametric external memory (Mai et al., 2023; Maharana et al., 2024; Yang et al., 2024a). This enhancement improves the agent's ability to explicitly store, retrieve, and utilize memory for various tasks. These works generally consist of two key stages: (1) recalling relevant thoughts from memory before generating a response, and (2) post-thinking after generating a response to incorporate both historical and new thoughts into memory, thereby improving consistency and efficiency.

In this subsection, we synthesize a wealth of research dedicated to leveraging external memory using refined and optimized RAG methods at inference time for the LLM agent on better downstream tasks.

### 4.1.1 Short-term Memory

Short-term memory serves as a transient storage system within LLMs, typically implemented by maintaining recent inputs within the context window. Due to the limited length of the context window, input history prior to a certain time point is inevitably discarded. Nevertheless, short-term memory enhances the continuity and consistency of LLMs and provides significant benefits in many challenging tasks, such as multi-hop reasoning and sequential decision-making.

**Chain-of-Thought (CoT)** (Wei et al., 2022) prompting explicitly asks LLMs to generate intermediate reasoning steps that lead to the final answer. These intermediate steps (also known as thoughts) are stored in the context of LLMs, acting as their short-term memory and stimulating logical reasoning. **COT-SC** (Wang et al., 2022b) is an extension of the CoT framework that introduces a decoding strategy called self-consistency to enhance complex reasoning performance in language models by sampling diverse reasoning paths and selecting the most consistent answer. Unlike CoT, where reasoning follows a linear-chain structure, **Tree of Thoughts (ToT)** (Yao et al., 2024a) prompting solves problems by considering multiple reasoning paths organized in a tree structure. This tree-structured reasoning facilitates efficient forward-looking and backward-tracking, which are crucial elements of advanced search-based problem-solving. Building on CoT and ToT, **Graph of Thoughts (GoT)** (Besta et al., 2024) prompting enables reasoning over a graph

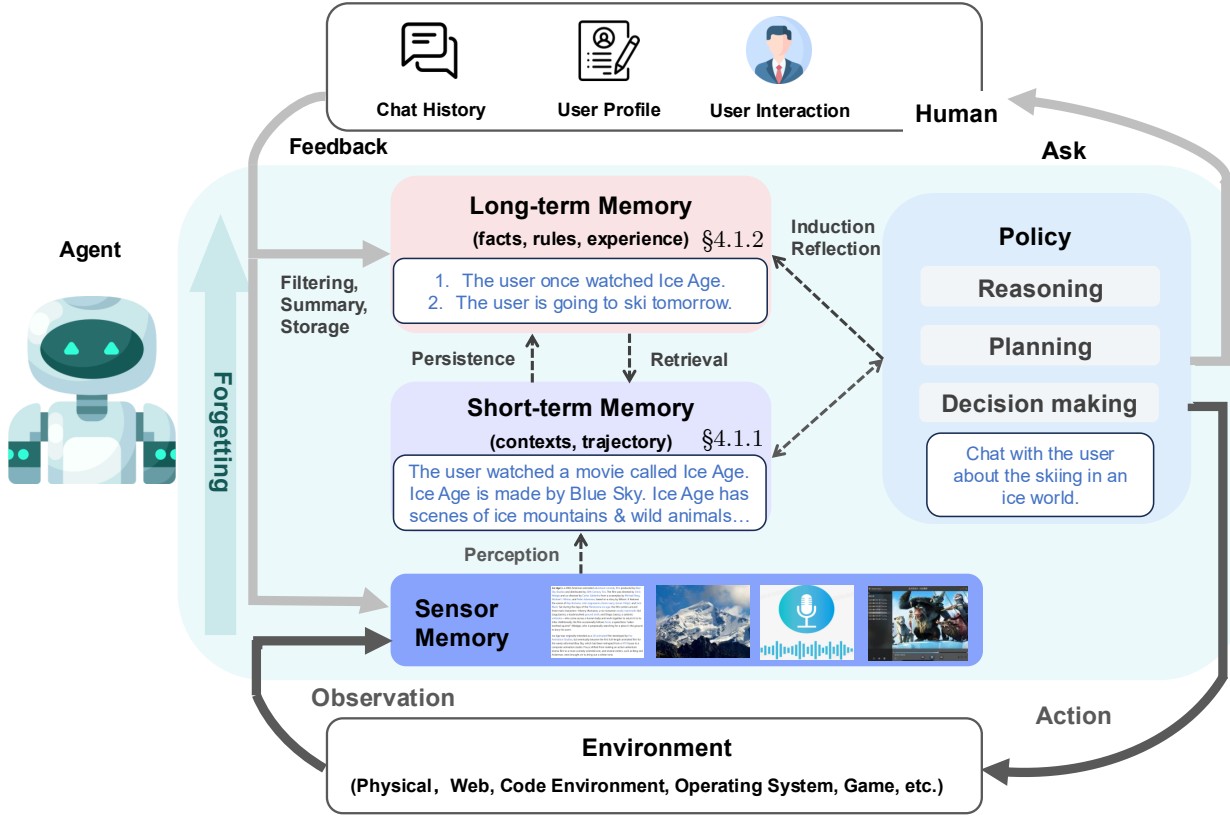

Figure 9: General memory architecture for a single agent. The agent interacts with the external environment (black arrows) and humans (gray arrows).

structure, where each node represents an intermediate thought, and the edges encode the dependencies between them. GoT offers a more flexible prompting paradigm, as it supports a wide range of thinking transformations. For example, it allows convenient aggregation of thoughts and seamless switching between different reasoning flows within the graph structure.

**ReAct** (Yao et al., 2022) is a novel approach that prompts LLMs to generate both reasoning traces and task-specific actions in an interleaved manner, fostering synergy between the two processes. This method combines reasoning and acting of LLM and allows the model to dynamically update action plans through reasoning and interact with external sources like Wikipedia to enhance decision-making. **Reflexion** (Shinn et al., 2024) builds upon ReAct and enhances the decision-making ability of LLMs using verbal reinforcement learning. At the core of Reflexion is deliberate reflection on natural language feedbacks obtained from task outcomes, rather than through weight updates. Reflexion firstly explores the property of self-reflection in LLMs and shows that self-reflection is extremely useful to iteratively learn over trials as another form of short-term memory. Similar to Reflexion, which leverages linguistic feedback, an increasing number of works have been proposed for self-improvement and evolution (Li et al., 2025b; Tang et al., 2024; Li et al., 2024b; Zhao et al., 2025). Gupta et al. (2024) learns general prompt instructions for LLMs using past self-reflections. Specifically, it gathers self-reflections in training and generalizes them into verbal 'meta-reflections' which serve as additional instructions to enhance the efficiency of agent's. RefAug (Zhang et al., 2024e) uses reflective augmentation to embed reflective sections within training instances, encouraging models to consider alternative solutions and engaging in deeper reasoning. Notably, This method goes beyond standard data augmentation by fostering a more comprehensive understanding of mathematical problems. Reflection on search Trees (RoT) (Hui et al., 2024) is introduced to reflect on an LLM's previous tree search experiences to generate guidelines, which are then used to improve the model's decisions in subsequent searches. This approach prevents repeated mistakes and enhances search efficiency. A key innovation is the identification

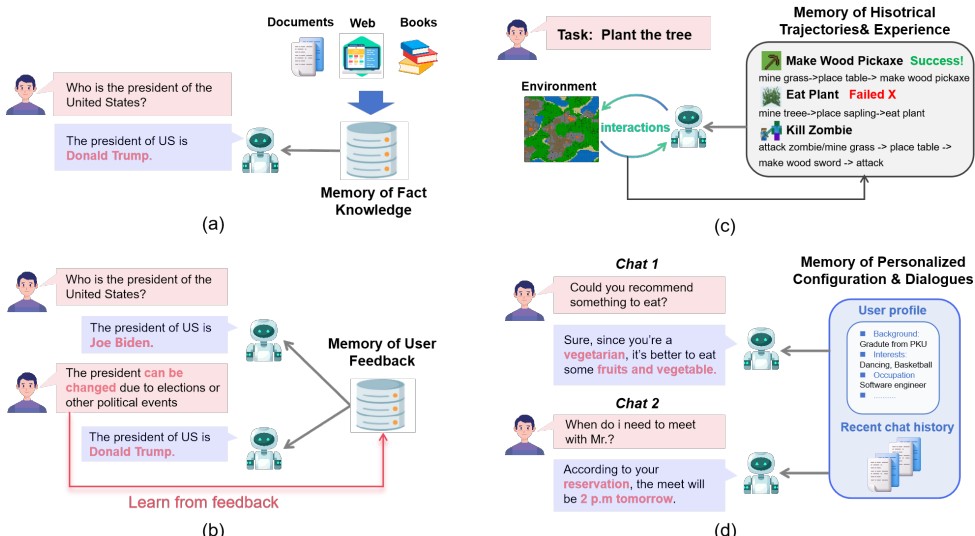

Figure 10: Long-term memory containing (a) fact knowledge; (b) historical trajectories and experience; (c) user feedback; (d) dialogues and personalized configuration.

of critical information from historical searches to produce more effective guidelines. Mirror (Yan et al., 2024) is a multiple-perspective self-reflection method to addresses the limitations of LLMs in self-assessment and feedback generation by introducing a Navigator-Reasoner framework. It improves through a heuristic interaction between a navigator, which provides question-adaptive directions, and a reasoner, which assesses and refines predictions based on the directions from the navigator. Textgrad (Yuksekgonul et al., 2024) is an automatic "differentiation" framework that optimizes composite AI systems by back-propagating textual feedback from large language models.

### 4.1.2 Long-term Memory

While operating in a real-time environment like Jia et al. (2024); Hafner (2021); Bellemare et al. (2013), crucial information, interactive feedback and distilled experience need to be continuously collected and further extracted from short-term memory and preserved as long-term memory of the LLM agent for future reference. To provide the agent with useful knowledge and experience for latter use, some works (Huang et al., 2024a; Park et al., 2023; Wang et al., 2023g; Zhang et al., 2024a) incorporate long-term memory (LTM) to LLMs from both user-specific and common-sense perspectives. This enables LLMs to flexibly utilize past experiences in accordance with current situations and enhance their task-planning and decision-making capabilities. These works usually formulate and organize thoughts in memory based on operations like insert, forget, merge and others, enabling dynamic updates and evolution of the memory in the long run.

As illustrated in Figure 10, this section focuses on four primary categories of long-term memory for an LLM agent:

**Memory of fact knowledge**  The vast number of parameters in LLM endows them with remarkable capabilities, allowing them to excel in a variety of Natural Language Processing (NLP) tasks. However, this complexity also presents challenges, making LLMs difficult to train and inhibiting their ability to continuously assimilate new knowledge(Zhang et al., 2023g), which may lead to inaccuracies in their outputs. In order to resolve these issues, Du et al. (2023) propose a continual learning framework that incorporates memory mechanisms allowing LLMs to assimilate new knowledge and modular operators to enhance model inference with this newly acquired knowledge and dynamically adapt to evolving environments and continuously integrate new information without the need for parameter tuning. Similarly, Modarressi et al. (2023) augments LLM with a write-read memory module by extracting and saving knowledge in the form of triplets,

allowing for scalable, updatable, interpretable, and aggregatable memory storage, which is particularly beneficial for handling temporal-based question answering tasks.

**Memory of historical trajectories and experience**   Incorporating memory into agents is crucial for enhancing their ability to remember historical trajectories, which ultimately improves decision-making and learning efficiency. By allowing agents to recall past experiences, they can identify patterns, make more informed predictions, and adapt their strategies based on historical data. This memory mechanism enables agents to build upon previous knowledge, facilitating more nuanced and contextually relevant responses. Delving into the integration of memory also supports the development of more sophisticated models that parallel the cognitive processes of humans, thereby advancing the overall effectiveness and versatility of artificial intelligence systems. Guo et al. (2023a) incorporate a centralized working memory hub and episodic buffer access to retain memories across episodes. This architecture aims to provide greater continuity for nuanced contextual reasoning in intricate tasks and collaborative scenarios. Liu et al. (2023a) also maintains an evolved memory for storing historical thoughts and employs Locality-Sensitive Hashing for efficient retrieval. Kagaya et al. (2024) introduces Retrieval-Augmented Planning (RAP) with a contextual memory module to leverage past experiences for improved decision-making in complex tasks. RAP dynamically retrieves relevant past experiences based on the current context to guide planning and action selection, mirroring human-like analogical reasoning. In recent, numerous studies have explored learning from past experiences through memory mechanisms, emphasizing the reuse of successful trajectories (Su et al., 2025) and procedural workflows (Fang et al., 2025a; Liu et al., 2025c).

**Memory for self-evolving**   More recently, to enhance agents' self-evolving capabilities, Liang et al. (2025a) proposed a memory optimization mechanism grounded in the Ebbinghaus forgetting curve and linguistic principles, aiming to emulate human-like memory processes within an agent's memory management system. Ouyang et al. (2025) introduced a reasoning memory framework, termed ReasoningBank. This framework is derived from past experiences and organizes them into structured knowledge units that abstract away low-level execution details while preserving transferable reasoning patterns and strategies. An agent equipped with the Reasoning Bank can leverage this curated repository of strategies to guide its decision-making, enabling it to recall effective insights, avoid previously encountered pitfalls, and adapt more robustly to novel queries.

**Memory of user feedback**   Recently, many researchers have proposed various methods for LLM to continue to improve without retraining. The Language Model (LM) is coupled with a growing memory module with feedback either corrections on the historical errors or better clarifications on the tasks from the users. Madaan et al. (2022b) maintains a growing memory of misinterpretation of user intents, along with user feedback for clarification. This memory enables the system to produce enhanced prompts for new queries based on past user feedback, effectively leveraging previous corrections to improve performance on similar tasks. Tandon et al. (2021) enables LLM's to improve their output after deployment without retraining, by leveraging user feedback. It maintains a dynamic memory of cases where users have identified and corrected output errors, and uses a trained corrector model to apply similar feedback to fix new errors. This approach shows significant improvement in repairing errors and avoiding past mistakes on new examples. The system represents a step towards continuous model enhancement through interactive learning and memory-based feedback reuse. Dalvi et al. (2022) integrates a dynamic memory component that stores user-provided corrections to the model's erroneous beliefs. These corrections are retrieved and used as additional context when answering new questions, helping the system avoid repeating past mistakes. This approach represents a novel application of memory-based continual learning for belief maintenance in language models, allowing for user-driven system enhancement over time. In order to maintain an ever-improving memory for LLM, Li et al. (2024b) utilizes recursive reasoning-based retrieval and experience reflections to continually update the memory and learn from communicative feedback provided by users. Without periodically re-training, it enables LLM to obtain fresh knowledge and historical experience by dynamically improving and growing a continually updated memory through human communications.

**Memory of dialogues and personalized configuration**   To address the limitation of context capacity over long conversations, Aadhithya A et al. (2024) introduces a novel memory structure that recursively

aggregates dialogue context flexibly to enhance long-term memory for dialogue agents. It allows for broad coverage of information with controlled depth through conditional tree traversals, balancing the breadth and depth of information for long-form dialogues, which is crucial for multi-turn reasoning without exponential parameter growth. Based on this, Chen et al. (2024a) adopts compressive memory that integrates session-specific summaries, user-bot dynamics, and past events into a concise memory format. This method is designed to be more manageable and efficient than traditional retrieval-based methods using Direct Preference Optimization (DPO) to enhance the model's ability to generate contextually appropriate responses. In Maharana et al. (2024), memory in these papers contains more nuanced and human-like conversational experiences, showing its effectiveness in managing time-dependent information and maintaining coherence in long-term and varied interactions, significantly contributing to the field of conversational agent.

Incorporating personalized knowledge bases into memory allows users to effectively store and access specific knowledge according to their requirements. Wang et al. (2023f) is a novel framework for knowledge retrieval and personalized knowledge base interaction. It employs the "Program of Thoughts" (PoT) prompting method, which facilitates model interaction with Knowledge Bases through the generation of Python code, thereby enabling knowledge retrieval. For domain specific tasks, Zhang et al. (2023c) introduces a personalized medical assistant tasks through a computational bionic memory. It utilizes a memory generation module using Dual-Process enhanced Memory (DPeM) that fine-tunes the LLM to produce personalized responses. Zhong et al. (2024) enables storing past conversations and adapting to user personalities. The system is showcased through SiliconFriend, a chatbot that provides empathetic and long-term companionship, demonstrating MemoryBank's effectiveness in improving AI engagement.

## 4.2 Multi-agent Memory

In LLM-based multi-agent collaboration, shared memory mechanisms enhance agents' ability to leverage historical information, improving reasoning and coordination. Prior works vary in focus, with some emphasizing real-time memory sharing for efficient reuse and others prioritizing agent autonomy by exchanging only essential information for flexibility and robustness, posing the challenge of balancing experience collection and information overload.

Conceptually, shared memory serves as a repository of past interactions and knowledge that agents query to inform current actions, often implemented as vector-based representations with similarity metrics retrieving relevant memories. Recent frameworks like A-MEM (Xu et al., 2025), DAMCS (Yang et al., 2025), MS (Gao & Zhang, 2024), and IoA (Chen et al., 2024b) offer innovative approaches to shared memory in multi-agent systems. Drawing inspiration from the Zettelkasten method, A-MEM emphasizes memory organization and evolution, creating interconnected knowledge networks through dynamic indexing and linking, where comprehensive notes with structured attributes evolve over time to refine contextual understanding and improve performance in long-term tasks and multi-hop reasoning. DAMCS enables decentralized cooperation through hierarchical knowledge graphs, dynamic team formation, and structured communication, allowing agents to learn from each other's experiences and adapt to new environments via most relevant information, while keep their own individual memories. MS focuses on real-time memory sharing by storing Prompt-Answer (PA) pairs in a shared memory pool, with an autonomous retriever ensuring relevant memories enhance response quality and reduce dependence on external databases. IoA adopts an Internet-inspired framework, enabling seamless integration of heterogeneous agents via an instant-messaging-like architecture, facilitating agent discovery, dynamic team formation, and structured communication for scalable collaboration.

## 4.3 System Architecture

To enable LLM with external memory capabilities, a variety of open-source tools and frameworks have been developed. As shown in Figure 11, these systems typically consist of the following modules: **Data Ingestion → Storage and Retrieval → User Interfaces and Application Invocation**. In this section, we compare the existing mainstream agent system architectures module by module. A summary of these comparisons is shown in Table 2. Notably, evaluation of the memory systems is also an important module, which is discussed in the next subsection.

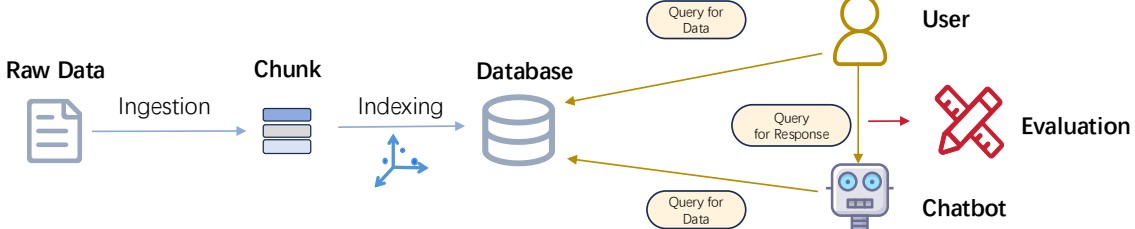

Figure 11: A general architecture of memory-augmented agent pipelines.

Table 2: Comparison of Tools for Data Ingestion, Storage, and User Interfaces

| Tool | Data Ingestion | Storage and Retrieval | User Interfaces |
|---|---|---|---|
| *Common* | • Universal connectors (files/web/DBs) | • Vector DB
• Hybrid indexing | • API access
• Logging tools |
| **MemGPT** | • Conversation context
• Tool execution outputs | • Hierarchical tiers | • Notebook examples |
| **Zep** | • Temporal streams
• Agent message graphs | • Graphiti engine
• Relationship expiration | • Dashboard monitoring
• Graph visualization |
| **Mem0** | • Cloud sync
• Cross-session data | • Managed backend
• Versioned storage | • Personalized UI
• Memory triggers |
| **Haystack** | • SQL/API ingestion
• Streaming pipelines | • Hybrid search | • REST API
• Pipeline configurator |
| **LangChain** | • Modular loaders
• Notion/Slack integration | • Conversation buffer | • Memory inspection
• Chain debugging |
| **LlamaIndex** | • 160+ formats
• Structured data parsing | • Tree-Index | • Query analyzers
• Index visualization |

### 4.3.1 Data Ingestion

**Comprehensive frameworks** such as LangChain[3], LlamaIndex[4], and Haystack[5] provide extensive data connectors that support a wide range of data sources. These include local files (e.g., TXT, PDF, Word), web scraping, database integration, APIs, and streaming data. For example, LlamaIndex natively supports over 160 data formats, covering plain text, tables, SQL databases, notion documents, Google Drive, Slack messages, and API responses, making it highly versatile for various business scenarios. These frameworks often allow customization of data preprocessing, such as specifying chunk size, filtering special symbols, and extracting metadata to optimize indexing efficiency.

**Conversational and agent memory** focus mainly on data generated from interactions. For instance, **Letta (MemGPT)**[6] emphasizes conversation context and tool usage results, with data sources including user messages, model responses, and outputs from external tools (e.g., code execution results). Text retrieved from a search tool may be incorporated as new memory. In multi-agent scenarios, data sources additionally include messages exchanged between agents.

**Domain-specific knowledge-based** applications, such as customer service and knowledge retrieval systems, rely on data sources such as product documentation, procedural manuals and FAQs. Tools like Haystack facilitate efficient document import, enabling bulk loading and automatic conversion into internal indexing formats.

---

[3]https://www.langchain.com/
[4]https://www.llamaindex.ai/
[5]https://haystack.deepset.ai/
[6]https://www.letta.com/

### 4.3.2 Storage and Retrieval

**Vector databases** are the most prevalent approach, which emphasizes large-scale semantic query performance. Many frameworks integrate vector libraries such as FAISS[7], Chroma[8], Pinecone[9], Weaviate[10], and Milvus[11] by default. LangChain and LlamaIndex use local vector storage (either in memory or on disk) to store embeddings and can optionally connect to external service-based vector databases for persistence and distributed capabilities. In vector-based storage, text is segmented and encoded in high-dimensional vectors using models like Sentence-BERT (Reimers & Gurevych, 2019). The index consists of these vectors, which are matched with query vectors during retrieval to return relevant results. Retrieval hyper-parameters, such as top-K value or similarity threshold, can be fine-tuned. Some tools employ additional strategies, such as summary or hierarchical indexes. LlamaIndex's Tree-Index organizes documents into a hierarchical structure, enabling efficient retrieval for long-document QA. Keyword or hash indexing can also complement vector searches to improve accuracy and speed.

**Graph databases** excel in structured knowledge representation, logical reasoning, relationship tracing, and version control. Systems like Zep[12] parse conversations into nodes and edges with attributes (e.g., context summaries, embeddings, timestamps). Its Graphiti engine dynamically updates the graph, invalidating outdated relationships. Retrieval utilizes graph algorithms and explores node relationships. Mainstream frameworks like LlamaIndex and LangChain also support graph databases for enhanced knowledge management.

**Hybrid storage** strategies are sometimes employed to meet diverse needs, balancing speed and capacity. For instance, Letta introduces hierarchical storage, keeping recent conversations in immediate context while compressing older information into external databases for long-term archiving. Persistence is another major consideration. If vector indices stored in memory are not saved, they are lost upon restart. Hence, most frameworks offer options for persisting indexes and memory, such as dumping indexes to disk files or directly to clouds. Mem0[13] provides cloud-hosted services, allowing developers to store memory in its managed backend for durability and cross-session sharing.

**Traditional databases and file storage** are also used in simpler cases, which may rely on key-value stores (e.g., Redis), relational databases, or local files to record full conversations. However, such approaches struggle to scale to complex scenarios and are gradually being replaced by vector databases.

**Information retrieval methods** vary by application needs: chatbots prioritize recent conversation context and related knowledge, often using sliding windows (e.g., LangChain's ConversationBufferWindowMemory), while knowledge QA systems focus on pinpointing accurate answers from large databases, favoring semantic search combined with cross-verification. The organization of retrieval outputs is crucial; frameworks often truncate or filter retrieved memories to fit within context windows, with methods allowing LLMs to generate summaries from multiple sources. Graph community summaries similarly aggregate node content into concise contexts, ensuring prompts contain essential information without exceeding length limits.

### 4.3.3 User Interfaces and Application Invocation

**Developer frameworks** like LangChain, LlamaIndex, and Haystack provide APIs and libraries for integration but lack graphical user interfaces (GUIs). Memory functions operate in the background, allowing developers to review retrieval results through logs or debugging tools. API-based services like Zep store and retrieve conversation history through API calls, operating invisibly to end-users. Developers monitor usage via dashboards that track stored conversations and vector indexes. Command-line tools and developer utilities are available in some open-source projects. For example, MemGPT provides Jupyter Notebook examples for API-based memory retrieval.

---

[7]https://faiss.ai/
[8]https://www.trychroma.com/
[9]https://www.pinecone.io/
[10]https://weaviate.io/
[11]https://milvus.io/
[12]https://www.getzep.com/
[13]https://mem0.ai/

Table 3: Characteristics of memory.

| Feature | Type | STM | LTM | Subjective eval | Objective eval | Description |
|---------|------|-----|-----|-----------------|----------------|-------------|
| Temporality | Direct | ✓ | ✓ | ✓ | ✗ | Whether the memory includes time mentions, timestamp, temporal-based long dependency correlations |
| Consistency | Direct | ✓ | ✓ | ✓ | ✗ | Whether the memory remains to be consistent or not |
| Redundancy | Direct | ✓ | ✓ | ✓ | ✗ | Whether the memory maintains redundant information or not |
| Variance | Direct | ✗ | ✓ | ✓ | ✗ | Whether the memory is static or dynamic that can be updated |
| Transformation | Direct | ✗ | ✓ | ✓ | ✗ | Whether the memory can be converted between short-term and long-term |

**Full-featured platforms** such as Dify[14] and Mem0 offer web-based UIs for chatbot configuration, knowledge base management, and real-time interactions. Dify enables non-programmers to build memory-enabled bots visually, while Mem0 provides a hosted ChatGPT with persistent memory, allowing users to upload knowledge and personalize interactions.

## 4.4 Evaluation on Agent Memory

### 4.4.1 Qualitative Evaluation

Understanding the characteristics of the agent memory is crucial for designing systems that effectively store, retrieve, and utilize information. In this subsection, we highlight the role of memory in optimizing performance, ensuring efficiency, and enabling communicative learning for an LLM agent. By examining these features, we aim to provide a comprehensive overview of memory's multifaceted nature and its impact on system behavior, as shown in Table 3:

**Temporality** indicates whether temporal facts or events exist in the memory and how they are stored, accessed, and utilized over time. Memory with time-sensitive information is crucial for tasks involving temporal understanding, reasoning and long dependency tracking, which enables the model to distinguish information in a timeline order. **Consistency** reflects whether the memory remains stable and consistent across interactions. It ensures the credibility and reliability of the LLM agent when generating the output based on relevant memory in context. In the meantime, it avoid frustration when retrieving conflicting information across different queries. **Redundancy** indicates whether the memory maintains redundant information, such as storing multiple versions of the same fact or event. While redundancy can serve as a backup for fault tolerance, excessive redundancy can lead to inefficiency and confusion. **Variance** refers to the memory being dynamic that it can be updated and merged with new incoming information without overwriting critical old data or losing coherence. Dynamic updating is critical for LLM agents in real-time applications, such as interactive agents or systems that must adapt to new facts or corrections. **Conversion & Transformation** refers to whether memory can be transferred between short-term memory (STM) and long-term memory (LTM) effectively. Real-time interactions benefit from STM, whereas LTM supports knowledge retrieval for tasks requiring continuity over sessions, comprehensive reasoning, and continuous learning.

### 4.4.2 Quantitative Evaluation

The quantitative evaluation of an agent's memory is imperative to validate design choices and direct future research. Such evaluations could be conducted across diverse tasks and levels of granularity, encompassing

---

[14]https://dify.ai/

both final task outcomes and the intrinsic properties of the memory module. Here we present a comprehensive framework of the evaluation landscape.

**Evaluation tasks.** The choice of task is crucial, as it defines the environment and the types of tasks in which the evaluated agents must utilize memory. A growing number of benchmarks have been developed to assess agents in diverse scenarios. These can be broadly categorized into three main types.

- **Long-text and question-answering tasks:** These benchmarks evaluate an agent's ability to retain, retrieve, and reason over information from long contexts, serving as a measure of long-term memory capabilities in static settings. Examples include: **NarrativeQA** (**Kočiský et al., 2018**), **QuALITY** (**Pang et al., 2021**) and **Loogle** (**Li et al., 2023a**), which require agents to read lengthy narratives and answer comprehension questions, thereby testing their ability to track long-range dependencies; **RetrievalQA** (**Zhang et al., 2024f**), a benchmark for open-domain question answering that assesses the model's capacity to retrieve relevant documents from a large corpus and synthesize answers; and **Needle In A Haystack (NIAH)** (**Kamradt, 2024**), a synthetic test designed to precisely measure an agent's ability to recall a specific fact embedded within a large volume of irrelevant context, directly probing the limits of memory recall.

- **Interactive and Open-World Tasks:** These benchmarks assess memory in dynamic, multi-step decision-making scenarios where the agent must interact with an environment. Success in these tasks strongly indicates effective memory use for planning, adaptation, and learning. Examples include: **ALFWorld** (**Shridhar et al., 2021**), a simulated home environment where agents execute multi-step natural language instructions to complete household tasks, requiring memory of both instructions and environmental state; **WebArena** (**Zhou et al., 2024b**), a realistic and complex benchmark in which agents perform tasks on live websites (e.g., booking a flight or making a purchase), demanding robust memory of past actions, website responses, and overall goals across multiple pages and interactions; **Gentopia** (**Xu et al., 2024**), an ecosystem designed for building and evaluating agents, offering a platform for standardized testing across various predefined tasks; and **AgentBench** (**Liu et al., 2023b**), a multi-dimensional benchmark that evaluates LLM-based agents across a diverse set of environments—including operating systems and games—testing both general reasoning and interactive abilities.

In these tasks, three levels of metrics can be considered: general-purpose task-oriented metrics that gauge overall performance, low-level metrics that probe the memory's fundamental capabilities and specialized metrics for communicative learning agents.

**General-Purpose Task-Oriented Metrics.** These are high-level, extrinsic metrics that measure the ultimate success and efficiency of an agent on a given benchmark. **Task Success and Accuracy** is the most critical metric, quantifying the percentage of tasks the agent successfully completes or the accuracy of its final answers. It serves as the primary indicator of the agent's overall capability. **Efficiency (Latency and Cost)** is also paramount in real-world applications. This metric is typically assessed through **inference latency** (the time taken to complete a task), the **number of steps or actions** required, and the associated **monetary cost** (e.g., the total tokens consumed via API calls).

**Low-Level Intrinsic Memory Metrics.** These fine-grained metrics assess the intrinsic properties of the memory module, providing deeper insights into its functionality beyond just task success. Wu et al. (2024a) proposes a detailed taxonomy of such capabilities, which we adapt and summarize here: **Information Extraction (IE):** The agent's ability to accurately recall previously seen information (recall rate) and extract specific facts from its memory without hallucination (precision). **Multi-Session Reasoning (MR)**: Ability to synthesize the information across multiple history sessions to answer complex questions that involve aggregation and comparison. **Knowledge Utilization (KU)**: Ability to recognize the changes in the user's personal information and update the knowledge of the user dynamically over time. **Temporal Reasoning (TR)**: Awareness of the temporal aspects of user information, including both explicit time mentions and timestamp metadata in the interactions. **Abstention (ABS)**: Ability to identify questions seeking unknown information, i.e., information not mentioned by the user in the interaction history, and answer "I don't know".

**Comparative Evaluation of Memory Frameworks on General and Low-Level Metrics**

Table 4: Performance evaluation with Llama3-8B-IT as the reasoning engine. Each cell shows correctness rate (bold) and average time in seconds (footnotesize).

| | | | | Llama3-8B-IT | | | |
|---|---|---|---|---|---|---|---|
| **Framework** | **KU** | **MS** | **SS-Assist** | **SS-Prefer** | **SS-User** | **TR** | **Overall** |
| No Memory | **0.000** | **0.000** | **0.000** | **0.000** | **0.000** | **0.000** | **0.000** |
| | 1.62 | 1.54 | 2.25 | 4.78 | 1.46 | 1.25 | 1.73 |
| ChromaDB | **0.625** | **0.074** | **0.900** | **0.333** | **0.857** | **0.444** | **0.470** |
| | 5.34 | 6.02 | 5.57 | 8.42 | 5.32 | 6.67 | 6.09 |
| Langchain | **0.026** | **0.000** | **0.036** | **0.000** | **0.000** | **0.023** | **0.032** |
| | 112.35 | 126.60 | 108.72 | 105.83 | 100.99 | 115.41 | 111.65 |
| Haystack | **0.562** | **0.111** | **0.800** | **0.167** | **0.857** | **0.741** | **0.530** |
| | 0.88 | 1.95 | 1.54 | 3.61 | 0.75 | 2.26 | 1.75 |
| LlamaIndex | **0.714** | **0.636** | **0.500** | **0.500** | **0.727** | **0.615** | **0.646** |
| | 25.47 | 29.35 | 25.77 | 28.61 | 26.40 | 27.62 | 27.31 |
| Mem0 | **0.583** | **0.500** | **0.625** | **0.567** | **0.500** | **0.592** | **0.555** |
| | 2280.15 | 1632.03 | 1956.42 | 2103.78 | 2927.99 | 1765.34 | 2110.95 |
| Zep | **0.125** | **0.148** | **0.364** | **0.200** | **0.143** | **0.259** | **0.200** |
| | 172.84 | 178.92 | 169.35 | 181.67 | 176.43 | 177.21 | 176.07 |

For our evaluation, we selected the `longmemeval_s_cleaned` dataset from the official LongMemEval benchmark Wu et al. (2024a). This benchmark is uniquely suited for our purposes as it combines the characteristics of the two critical evaluation tasks: Long-Context Question Answering (QA) and interactive tasks derived from real user conversations. This allows us to measure not only the accuracy of QA responses but also the processing time for data ingestion, retrieval, and reasoning. The LongMemEval dataset categorizes test cases into several low-level types based on the required memory capabilities: Knowledge Updates (KU), Multi-Session Reasoning (MR), Single-Session Reasoning (SS-Assistant), Single-Session Reasoning (SS-Preference), Single-Session Reasoning (SS-User), and Temporal Reasoning (TR). The full dataset comprises 2,500 questions.

We benchmarked a variety of memory frameworks, including a baseline with no memory, a simple RAG implementation using ChromaDB, Langchain's native FAISS RAG, Haystack, LlamaIndex, Mem0 (local version), and Zep (API version). Due to the extensive average processing times of Mem0, Langchain, and Zep, we conducted their evaluations on a 10% random sample of the dataset. For the reasoning engine, we employed two LLMs: Llama-3-8B-IT and GPT-4o-mini. All responses were evaluated for correctness using GPT-4o-mini.

The results, presented in Table 4 and Table 5, show the correctness rate and the average total time per question (in seconds), which includes data ingestion, retrieval, and reasoning. For all frameworks except the no-memory baseline, ingestion and retrieval constituted the majority of the processing time.

Our findings indicate that the simplest framework, ChromaDB, performed surprisingly well. In contrast, many of the more complex frameworks did not deliver the performance improvements their official documentation suggested, at least on this comprehensive task. Langchain's memory implementation was the least effective, appearing largely non-functional. Mem0 exhibited extremely long processing times, as it performs significant internal memory organization and inference during each data ingestion, which did not translate into a proportional increase in accuracy. Zep also showed prolonged data ingestion times and the ob-

Table 5: Performance evaluation with GPT-4o-mini as the reasoning engine. Each cell shows correctness rate (bold) and average time in seconds (footnotesize).

| | GPT-4o-mini | | | | | | |
|---|---|---|---|---|---|---|---|
| Framework | KU | MS | SS-Assist | SS-Prefer | SS-User | TR | Overall |
| No Memory | **0.060** 1.27 | **0.000** 1.04 | **0.000** 1.18 | **0.000** 1.93 | **0.000** 0.92 | **0.000** 1.15 | **0.010** 1.16 |
| ChromaDB | **0.752** 5.88 | **0.222** 6.23 | **1.000** 6.79 | **0.833** 7.58 | **0.865** 6.15 | **0.563** 6.65 | **0.600** 6.41 |
| Langchain | **0.031** 108.45 | **0.000** 126.60 | **0.022** 103.78 | **0.000** 98.63 | **0.000** 100.99 | **0.019** 112.92 | **0.022** 108.56 |
| Haystack | **0.688** 1.00 | **0.148** 1.58 | **0.900** 0.99 | **0.500** 2.98 | **0.929** 1.55 | **0.852** 1.71 | **0.630** 1.00 |
| LlamaIndex | **0.712** 31.70 | **0.642** 27.28 | **1.000** 26.74 | **0.500** 35.35 | **0.823** 26.34 | **0.467** 28.28 | **0.667** 28.34 |
| Mem0 | **0.580** 1980.25 | **0.624** 2015.73 | **0.592** 1992.48 | **0.608** 2008.16 | **0.573** 1975.92 | **0.631** 2023.67 | **0.602** 2106.53 |
| Zep | **0.500** 178.45 | **0.270** 172.83 | **1.000** 181.29 | **0.500** 176.92 | **0.830** 173.67 | **0.550** 175.30 | **0.550** 176.41 |

served performance metrics demonstrated some variance from those claimed in the original research Rasmussen et al. (2025). Haystack and LlamaIndex, however, demonstrated a strong balance of performance and efficiency.

Across all frameworks, the multi-session reasoning tasks proved to be the most challenging. This difficulty is attributable to the large volume of conversational history that must be comprehensively processed, with each question averaging 47 sessions, each containing around 10 conversational turns. We attribute ChromaDB's strong performance in part to its effectiveness on single-session tasks, where relevant information can be extracted from a single contiguous block of memory. In comparison, the more advanced frameworks, which actively organize and synthesize memory, showed a slight advantage in the more demanding multi-session tasks.

**Specialized Metrics for Communicative Learning Agents.** We propose two types of test type to evaluate the capability of an agent memory as below:

- **Functionality Test (FT)** is a type of black-box testing that evaluates whether the software system performs its intended functions correctly as expected according to the defined requirements. It focuses on verifying the functional correctness of the application by checking specific features and operations about the memories. Here we use FT to test capabilities like *Learning Efficiency* and *Generalization.*

- **Perturbation Test (PBT)** evaluates the stability of a software system by introducing modifications or disturbances to its inputs, environment, or internal state. The goal is to discern how the system reacts to perturbations of the memories and whether it can maintain the expected behavior. Here we use PBT to test capabilities like *Controllability* and *Robustness.*

The detailed definitions of the evaluation metrics mentioned above can be found in Table 6. We conducted these tests based on RetrievalQA Zhang et al. (2024f) dataset using four different types of agents, all using Llama3-8B-IT(Grattafiori et al., 2024) as the base model, with varying memory settings:

Table 6: Memory capabilities metrics.

| Test type | Capability | Type | STM | LTM | Sub. eval | Obj. eval | Description |
|---|---|---|---|---|---|---|---|
| **FT** | Learning Efficiency | Indirect | ✗ | ✓ | ✗ | ✓ | Task performance increase with accumulative memory |
| | Generalization | Indirect | ✗ | ✓ | ✗ | ✓ | Unseen task performance with the memory learnt from historical tasks |
| **PBT** | Controllability | Indirect | ✓ | ✓ | ✗ | ✓ | Task performance provided with unknown or counterfactual contexts |
| | Robustness | Indirect | ✓ | ✓ | ✗ | ✓ | Task performance provided with irrelevant contexts as noise |

Table 7: Evaluations on capabilities of memory using different agents. $M^{upd}$ indicates that the ground truth of the current question will be updated into the memory.

| | ICL | | RAG | | RAM | | Concordia Agent | |
|---|---|---|---|---|---|---|---|---|
| | **wo** $M^{upd}$ | **w** $M^{upd}$ | **wo** $M^{upd}$ | **w** $M^{upd}$ | **wo** $M^{upd}$ | **w** $M^{upd}$ | **wo** $M^{upd}$ | **w** $M^{upd}$ |
| Generalization | 20% | 20% | 38% | 38% | 52% | 52% | 16% | 16% |
| Controllability | 20% | 16% | 54% | 48% | 38% | 44% | 16% | 24% |
| Robustness | 20% | 4% | 38% | 19% | 54% | 26% | 16% | 12% |

- In-Context Learning (ICL): Memories are entirely stored in the LLM's context.

- Retrieval-Augmented Generation (RAG): Standard RAG for storing memories.

- RAM: Continuously improving memory based on external feedback, implemented according to Li et al. (2024b).

- General Agent: An agent that automatically decides how to handle external feedback at each step, implemented using the Concordia framework (Vezhnevets et al., 2023).

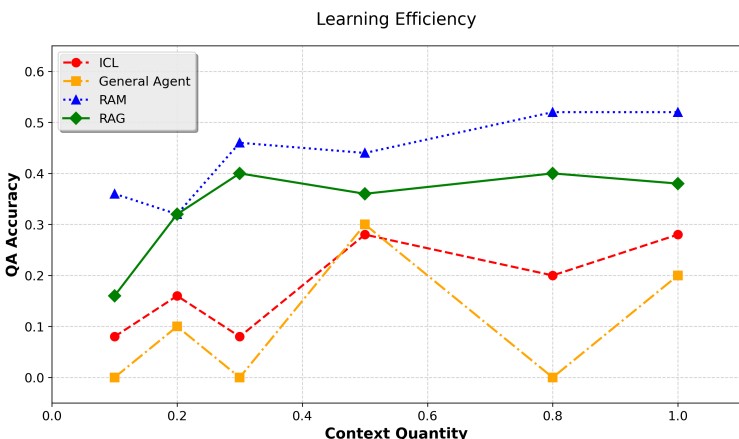

Figure 12: Learning Efficiency: Question-answering accuracy as a function of the proportion of necessary context provided to the LLMs.

In Figure 12, we illustrate the performance of each memory setting across varying context quantities, ranging from 10% to 100% of the memories required to answer the question. This analysis highlights the learning efficiency of RAG and RAM, as performance basically improves consistently as the proportion of necessary memories increases.

Table 7 summarizes the evaluation results for Generalization, Controllability, and Robustness. The applied memory perturbations significantly influence performance, demonstrating the sensitivity of these metrics to changes in memory inputs. However, the generalization ability of past memories has not been observed, potentially due to insufficient data volume.

### 4.5 Limitations, Open Questions, Discussion

In contrast to Zhang et al. (2024d), which posits that memory is the definitive component that elevates a standard LLM into an autonomous, self-evolving agent and focuses on functional, examining memory through the lens of what is required to build effective agents, our discussion of agentic memory concentrates on the design and implementation of memory systems developed for direct deployment in scenarios where memory mechanisms are indispensable. Likewise, in distinction to Du et al. (2025), who argues for a more fundamental, mechanistic understanding and contends that previous application focused reviews have overlooked the "atomic operations", such as consolidation, retrieval, and forgetting, that form the universal building blocks of any memory system, regardless of its application, our survey presents the broadest perspective, framing memory as the foundational element augmenting modern LLMs and, critically, Multi-Modal LLMs, encompassing the challenges of integrating information across diverse modalities like text, vision, and audio, thereby offering a generalizable system view for memory in Agents.

To conclude, to enhance the utility of agents, enabling dynamic memory adaptation during reasoning and communicative learning could be crucial. Inspired by human pedagogy, methods like RAM (Li et al., 2024b) demonstrate the potential of recursive retrieval and experience reflection for continuous memory updates based on user feedback. Additionally, in multi-agent systems, ensuring adaptive network structures and robust communication frameworks (Mao et al., 2024; Marro et al., 2024; Liu et al., 2024b) to facilitate effective memory synchronization also remains as a challenge.

## 5 Memory-augmented Multi-Modal Large Language models

Addressing the complexities inherent in multimodal context modeling, a critical research inquiry emerges:

*How can we devise and execute memory mechanisms that adeptly amalgamate and preserve extensive multimodal contextual data, thereby augmenting the comprehension and manipulation of intricate datasets within fluid environments?*

This question is particularly salient within the domains of vision and robotics, where the optimization of contextual memory is paramount for bolstering the cognitive and functional capacities of embodied agents. Such advancements would empower these agents to execute sophisticated operations, enabling systems to make informed decisions based on comprehensive contextual understanding.

### 5.1 Multimodal Context Modeling with Memory

In this section, we introduce the multimodal context modeling with memory, incorporating information from audio, video, and other modalities. For each modality, we will discuss its modeling in relation to various downstream tasks.

#### 5.1.1 Audio Context Modeling

The continuous and high-frequency nature of audio presents significant challenges in efficiently modeling its sequences, demanding substantial computational resources. As a result, developing effective methods to incorporate audio history is crucial for various applications. Recent advancements in audio context modeling have introduced innovative solutions to address these challenges. For instance, Conformer-NTM (Carvalho & Abad, 2023) proposes an external memory network between the encoder and decoder transformers for automatic speech recognition (ASR), enhancing the system's ability to handle complex audio sequences. Similarly, Loop-Copilot (Zhang et al., 2023d) introduces a Global Attribute Table that identifies and manages various musical attributes at any moment, aiding in task execution and ensuring musical coherence in music generation. Furthermore, MR-MT3 (Tan et al., 2024) employs previous instrumental tokens as key/value

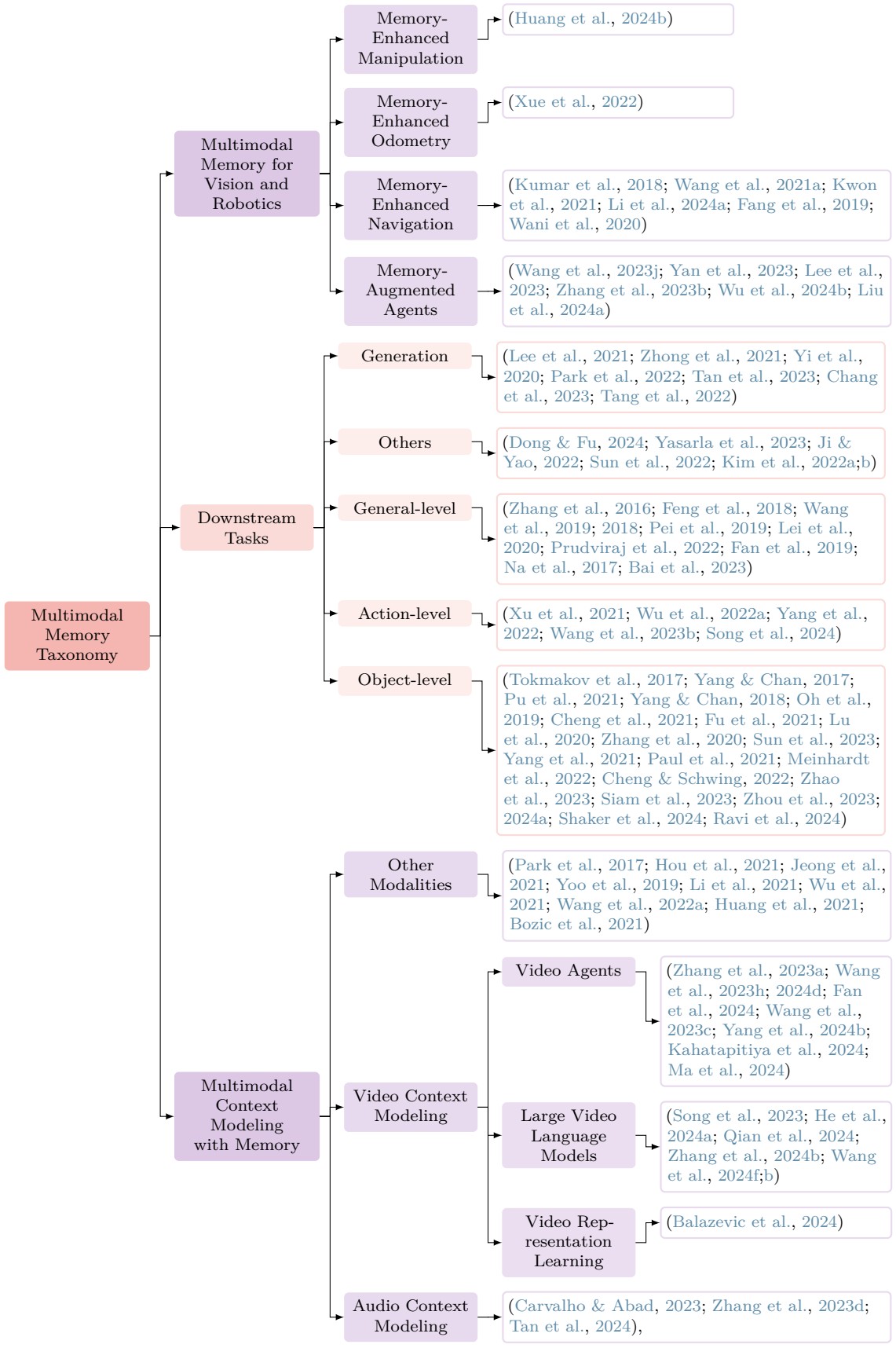

Figure 13: Taxonomy of multimodal memory applications and downstream tasks.

memory, effectively preventing instrument leakage in automatic music transcription tasks. These approaches demonstrate promising strategies for improving audio modeling, paving the way for more efficient and coherent audio processing systems.

### 5.1.2 Video Context Modeling

Video context modeling presents unique challenges compared to image processing due to its added time dimension, which significantly increases complexity. Most existing approaches use sampling-based methods, converting continuous video into discrete stacked frames. This makes balancing computational complexity with detailed video information a crucial focus in video research. Memory mechanisms have emerged as a vital strategy in achieving this balance. This section explores various aspects of video context modeling, starting with general video representation learning, focusing on developing more effective video encoders. We then delve into recent advancements in memory mechanisms for large video language models and video agents, which demonstrate strong performance in zero-shot video-language benchmarks and applications. Finally, we discuss the role of memory in various downstream tasks.

**Memory-enhanced Video Representation Learning** In the realm of video representation, MC-ViT (Balazevic et al., 2024) introduces a long video encoder using memory consolidation and cross-attention on video segments to efficiently encode long-context videos. This enhances the ability to handle extensive video data while maintaining detailed representation.

**Large Memory-enhanced Video Language Models** For large video language models, several innovative memory-augmented approaches have been developed for long video modeling. MovieChat (Song et al., 2023) builds on Q-Former for visual feature extraction with memory consolidation to model long videos for video question answering (QA). MA-LMM (He et al., 2024a) extends this with a retrieval strategy based on the semantic similarity of frame features. VideoStreaming (Qian et al., 2024) proposes an method combined with an adaptive memory selection strategy on the recurrent image feature, selecting a constant number of question-related memories using Gumbel-Softmax. Flash-VStream (Zhang et al., 2024b) presents a hierarchical memory system, incorporating FIFO queue for spatial features, and uses abstract memory implemented by cross-attention for whole video modeling. VideoLLaMB(Wang et al., 2024f) introduces a recurrent memory bridge with a memory cache to model video history in memory for long video understanding. Additionally, OmniDrive (Wang et al., 2024b) utilizes a memory bank for frames in autonomous driving-related QA.

**Memory-enhanced Video Agent** Video agents have made significant strides by transforming various video elements into text, such as captions, object names, and timestamps, which are then stored as external memory for LLMs to enhance video processing tasks. LLoVi (Zhang et al., 2023a), LifelongMemory (Wang et al., 2023h) and VideoAgent(Wang et al., 2024d) leverage captions as a memory for LLMs. Furthermore, VideoAgent (Fan et al., 2024) combine captioning, tracking, VQA modules to inject video object detection ability into LLM. ChatVideo (Wang et al., 2023c) utilizes captioning, tracking, and audio modules to extract information from video as memory. DoraemonGPT (Yang et al., 2024b) adopts a more comprehensive method by incorporating captioning, tracking, ASR, detection, action, and segmentation module to provide additional information for LLMs in video QA and segmentation tasks. LangRepo (Kahatapitiya et al., 2024) integrates captions, and timestamps as memory to LLMs for sequential understanding. Finally, DrVideo (Ma et al., 2024) reinterprets long-video understanding as a long-document comprehension task, effectively utilizing the power of large language models. These advancements underscore the diverse strategies employed to address the complexities of video context modeling and suggest promising directions for future research.

### 5.1.3 Other Modalities

Beyond audio and video context modeling, memory strategies are increasingly leveraged across various modalities to enhance context understanding and improve performance. In image captioning, the CSMN model (Park et al., 2017) enhances memory networks by using them as repositories for multiple types of context information, appending previously generated words to capture long-term information, and employing a CNN memory structure for better context representation. For anomaly detection, the DAAC model (Hou et al., 2021) modulates reconstruction capabilities by generalizing the memory module in a blockwise manner

using a multi-scale approach. In image-to-image translation, MGUIT (Jeong et al., 2021) explores memory networks to improve translation results. MemoPainter (Yoo et al., 2019) introduces a memory-augmented colorization model that achieves high-quality colorization with limited data. For blind face restoration, RMM (Li et al., 2021) proposes a wavelet memory module that stores spatial features of low-quality images and guides high-quality restoration. In semantic segmentation, a memory-based approach (Jin et al., 2021) stores significant training image representations, while MM-Net (Wu et al., 2021) uses learnable memory embeddings for few-shot segmentation, and CDFSS (Wang et al., 2022a) employs a meta-memory bank to bridge domain gaps. In deraining, MOSS (Huang et al., 2021) uses a self-supervised memory module to record prototypical rain patterns. Lastly, for 3D scene reconstruction, TransformerFusion (Bozic et al., 2021) proposes a hierarchical memory of input frame features for online video 3D scene reconstruction. These applications underscore the versatility and effectiveness of memory networks in improving context modeling across diverse tasks.

## 5.2 Downstream tasks

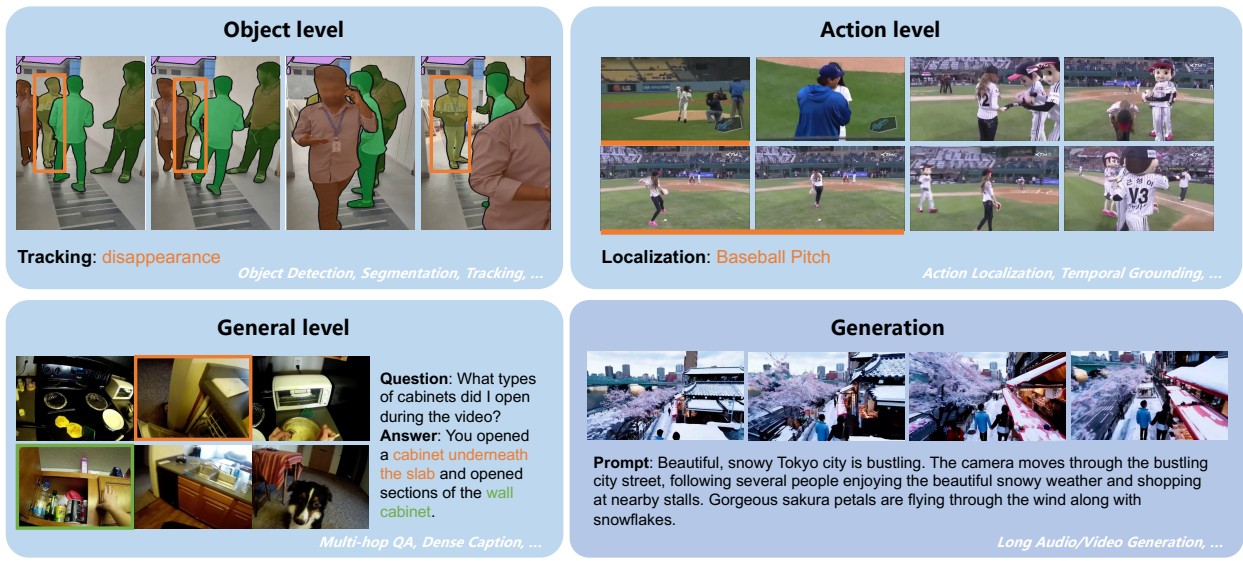

Figure 14: Various video understanding and processing tasks.

Advancements in video understanding and generation have highlighted the critical role of memory mechanisms in enhancing long-context modeling capabilities. Researchers have been increasingly focused on developing innovative approaches to leverage these memory mechanisms across various video processing tasks. This survey provides an overview of these advancements, beginning with video understanding at the object level, progressing through action-level tasks, and culminating in general-level applications such as summarization, captioning, and question answering. Figure 14 provides some tasks demonstration. Additionally, we explore recent breakthroughs in video generation that utilize memory-based architectures.

**Object-level Task**   Object segmentation and tracking in videos are foundational tasks that benefit significantly from memory-augmented models. Early efforts in this domain employed RNNs, LSTMs, and GRUs to maintain and update memory states over time. For instance, ConvGRU (Tokmakov et al., 2017) integrates convolutional gated recurrent units to track the evolution of objects within a scene, while RFL (Yang & Chan, 2017) utilizes convolutional LSTMs for object tracking with preserving video instory. RMAN (Pu et al., 2021) builds on the LSTM architecture by adding a memory activation layer specifically for visual tracking.

Further advancements introduced more sophisticated memory networks. MemTrack (Yang & Chan, 2018) employs a dynamic memory network to store and recall target information, utilizing an LSTM to manage memory operations for template-matching tasks. STM (Oh et al., 2019) and its enhanced version, STCN (Cheng et al., 2021), compute spatio-temporal attention across video frames, improving pixel-level object

distinction. STMTrack (Fu et al., 2021) introduces a mechanism that stores historical target information to guide the tracker toward the most informative regions.

Graph-based memory networks have also shown promise in video object segmentation. GraphMemVOS (Lu et al., 2020) leverages an episodic memory network structured as a fully connected graph, facilitating cross-frame correlation capture. DTMNet (Zhang et al., 2020) builds on this by incorporating both short- and long-term memory storage to enhance temporal modeling. TMRN (Sun et al., 2023) improves memory retrieval operations by spatially aligning memory frames with current frames before temporal aggregation.

The advent of transformer-based architectures has further revolutionized object segmentation and tracking. AOT (Yang et al., 2021) employs long-short term memory for associating multiple object segments, while IMANet (Paul et al., 2021) utilizes attention mechanisms to access semantic information stored in memory. TrackFormer (Meinhardt et al., 2022) and XMem (Cheng & Schwing, 2022) exemplify the integration of transformers with memory modules to handle occlusions and segment long video sequences effectively. Recent innovations continue to push the boundaries of memory utilization. S-ViT (Zhao et al., 2023) and MMC (Siam et al., 2023) focus on preserving detailed feature maps and reducing background confusion through multiscale memory transformers. RFGM (Zhou et al., 2023) introduces a relevance attention mechanism to adaptively assist in selecting pertinent historical information. RMem (Zhou et al., 2024a) enhances efficiency by restricting memory banks to essential frames. MAVOS (Shaker et al., 2024) optimizes long-term memory usage to ensure temporal smoothness without frequent expansions, and SAM 2 (Ravi et al., 2024) extends memory capabilities with a FIFO queue for seamless object tracking.

**Action level Task** Action classification and localization in videos are critical components of understanding dynamic scenes and require sophisticated models capable of identifying and interpreting temporal patterns. Recent advancements in this field have leveraged memory networks and transformer-based architectures to enhance the accuracy and efficiency of these tasks. One approach that has gained prominence is the use of memory networks. For instance, (Yuan et al., 2019) introduces a novel framework that writes significant information into an external memory module while discarding irrelevant data. This selective memory management improves video action recognition by focusing on key temporal elements. Transformer-based models have also been instrumental in advancing action classification and localization. LSTR (Xu et al., 2021) utilizes both long-term and short-term memory in a FIFO structure to address online action detection, allowing the model to maintain relevant past information efficiently. MeMViT (Wu et al., 2022a) proposes a hierarchical memory transformer that optimizes long-term memory use for effective video action classification and anticipation. Another significant contribution is RViT (Yang et al., 2022), which employs an attention gate to facilitate interaction between the current frame input and the previous hidden state. This mechanism enhances the model's ability to integrate past and present information dynamically. Similarly, MAT (Wang et al., 2023b) introduces a memory encoder that compresses both long-term and short-term memory in a segment-based manner. It also features a memory-anticipation circular decoder that updates historical and future representations for online action detection and anticipation. Finally, MATR (Song et al., 2024) presents a FIFO memory queue that selectively retains past segment features, optimizing the process of temporal action localization. This approach ensures that the most relevant temporal features are preserved, improving the model's ability to localize actions accurately over time.

**General level Task** Video summarization has evolved significantly with the introduction of memory-augmented models and deep learning architectures. Early approaches, such as vsLSTM (Zhang et al., 2016), employed LSTM to capture variable-range temporal dependencies among video frames. This method aimed to create both representative and compact video summaries by effectively modeling the temporal relationships inherent in video data. Building on this foundation, MAVS (Feng et al., 2018) introduced a memory-augmented extractive video summarizer that utilized an external memory to store comprehensive visual information from the entire video, enhancing the summarization process with high-capacity memory storage. Furthermore, SMN (Wang et al., 2019) stacked multiple LSTM and memory layers hierarchically. This approach integrated learned representations from prior layers, resulting in more precise video summaries for individual frames by capturing intricate temporal patterns.

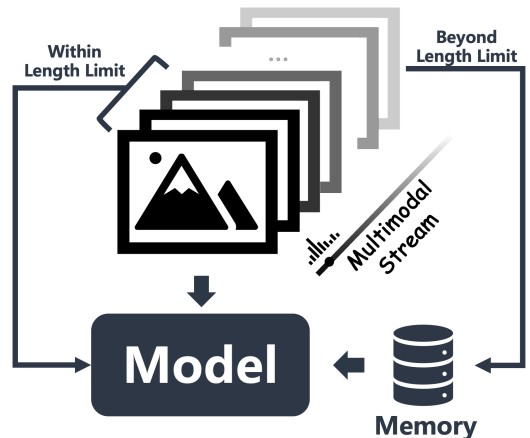

Figure 15: Memory for multimodal context modeling

Video captioning has also benefited from advancements in memory-augmented networks. M3 (Wang et al., 2018) proposed attaching an LSTM with an external memory that could store and retrieve both visual and textual content. This method allowed for multiple read and write operations, facilitating rich interactions between video sequences and corresponding sentences. MARN (Pei et al., 2019) introduced the Memory-Attended Recurrent Network, designed to explore the full-spectrum correspondence between words and their visual contexts, enhancing the captioning capabilities by leveraging memory structures. In terms of transformer-based solutions, MART (Lei et al., 2020) utilized a layer-wise TransformerXL architecture for video captioning, which was further improved by AAP-MIT (Prudviraj et al., 2022) through the integration of a Pyramid network for generating multi-sentence video descriptions, thereby enriching the narrative depth of the captions.

Video QA tasks have seen significant enhancements through the incorporation of sophisticated memory mechanisms and attention models. The Heterogeneous Memory Enhanced Multimodal Attention Model (Fan et al., 2019) introduced a heterogeneous external memory module on LSTM. This model employed attentional read and write operations to integrate motion and appearance features, co-learning attention mechanisms, and utilizing visual-question interactions to derive global context-aware representations. (Cai et al., 2020) further advanced Video QA by introducing fine-grained feature-augmented memories. This approach strengthened the information augmentation of video and text, improving memory capacity by capturing global interactions between high-level semantic information through self-attention and co-attention modules. RWMN (Na et al., 2017) utilized multi-layered CNNs to read and write sequential memory cells as chunks, effectively representing sequential stories with strong inter-block correlations. Lastly, Glance-Focus (Bai et al., 2023) proposed a two-stage method for Video QA. In the "glance" stage, an Encoder-Decoder generated dynamic event memories without supervision, while in the "focus" stage, these memories bridged the correlation between questions and both high-level event concepts and low-level video content, enhancing the model's comprehension and response accuracy.

**Other Understaning Task**   In addition to common video understanding tasks, memory-augmented models have been increasingly applied to various other video processing tasks, including flow estimation, depth estimation, video deblurring, gesture recognition, and visual speech recognition. Flow estimation has benefited from models like MemFlow (Dong & Fu, 2024), which employs memory storage for real-time flow estimation, retaining motion information to enhance accuracy. In depth estimation, MAMo (Yasarla et al., 2023) augments networks with memory to store learned visual and displacement tokens from previous frames, allowing for more accurate depth predictions through cross-referencing past features. Video deblurring has advanced with MmDeblur (Ji & Yao, 2022), which uses a memory branch to memorize blurry-sharp feature pairs, aiding the deblurring process for incoming frames. Gesture recognition has been improved by MENet (Sun et al., 2022), which features a dual-branch architecture to capture temporal dynamics between spatiotemporal windows. Visual speech recognition has seen enhancements through frameworks using associative bridges to

learn interrelationships and obtain target modal representations from memory (Kim et al., 2021), with VAM (Kim et al., 2022a) imprinting audio features into a memory network using visual features, and MVM (Kim et al., 2022b) employing multihead key memories for visual features and a value memory for audio knowledge to distinguish homophenes. These applications show the versatility of memory networks in enhancing video understanding systems by leveraging past information for improved accuracy and robustness in complex video analysis scenarios.

**Generation Task**    In the realm of video generation, there is a growing demand for creating long, high-quality videos. This challenge has led to the development of various memory-augmented models that enhance the quality and coherence of generated video content by leveraging advanced memory.

In the realm of video generation, there is a growing demand for creating long, high-quality videos, leading to the development of various memory-augmented models that enhance video content by leveraging advanced memory networks. The LMC-Memory model (Lee et al., 2021) utilizes memory alignment learning to store long-term motion contexts, improving video prediction by matching these contexts with sequences that exhibit limited dynamics. Similarly, MV-TON (Zhong et al., 2021) introduces a memory refinement module that embeds generated frames into a latent space as external memory, aiding subsequent frame generation with richer context. For talking face video generation, MemGAN (Yi et al., 2020) incorporates a memory-augmented GAN module to refine roughly rendered frames into realistic ones, enhancing video quality. Building on this, SyncTalkFace (Park et al., 2022) introduces an Audio-Lip Memory mechanism to align visual information of the mouth region with input audio, ensuring fine-grained audio-visual coherence. The EMMN model (Tan et al., 2023) constructs a Motion Memory Net that stores emotion embeddings and mouth motion features as key-value pairs, ensuring consistency between expression and lip motion. STAM (Chang et al., 2023) enhances spatiotemporal memorizing capacity by using a SpatioTemporal Attention based Memory on 3D-CNN, incorporating global spatiotemporal information to improve video prediction. Lastly, MemFace (Tang et al., 2022) addresses missing information in video generation with implicit and explicit memory components, capturing high-level semantics in the audio-expression shared space and aiding the neural-rendering model in synthesizing pixel-level details. These innovations underscore the critical role of memory networks in advancing video generation technology, enabling coherent and visually appealing outputs by effectively integrating past information and context.

## 5.3   Multimodal Contextual Memory for Robotics

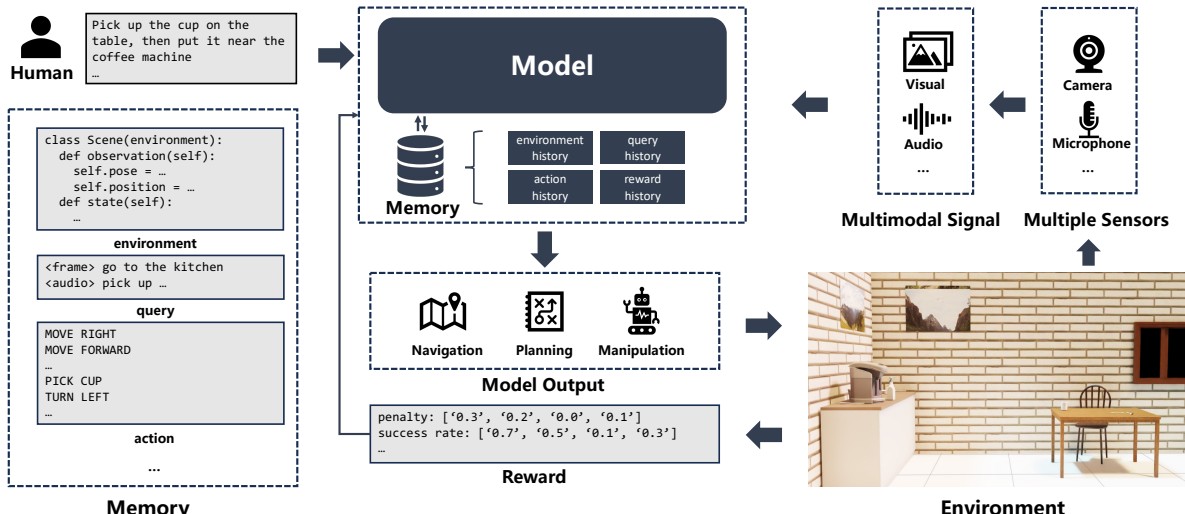

Figure 16: Memory for robotics

Integrating memory mechanisms into the capabilities of embodied agents and robotics has become increasingly essential for enhancing long-term planning, decision-making, visual navigation, and manipulation reasoning. These advancements demonstrate how memory can significantly improve an agent's ability to operate in complex environments.

### 5.3.1 Multimodal Memory-Augmented Agents

Multimodal memory-augmented agents demonstrate the integration of memory in diverse environments. JARVIS-1 (Wang et al., 2023j) equips an agent to perceive multimodal inputs, generate complex plans, and perform embodied control in a gaming environment like Minecraft, using memory to combine pre-trained knowledge with actual game experiences. MM-Navigator (Yan et al., 2023) employs GPT-4V for multimodal self-summarization in smartphone GUI navigation tasks, converting historical actions into concise natural language memory. MobileGPT (Lee et al., 2023) and AppAgent (Zhang et al., 2023b) focus on smartphone applications, accumulating knowledge about apps in graph form and summarizing interaction histories for improved decision-making and interpretability. OS-Copilot (Wu et al., 2024b) features a framework for building generalist agents capable of interacting with various elements in an operating system, using a configurator with working, declarative, and procedural memory. MEIA (Liu et al., 2024a) offers an embodied agent for cafe scenes, utilizing a multimodal environment memory module that stores key scene information in natural language, guiding large models to execute action plans effectively under diverse requirements.

### 5.3.2 Memory-Enhanced Navigation, Odometry, and Manipulation

In the field of visual navigation, memory mechanisms play a crucial role in enabling agents to navigate effectively. RPF (Kumar et al., 2018) abstracts sequences of images and actions into memories for robust path following using RNNs. SSM (Wang et al., 2021a) introduces an external structured memory that stores visual and geometric information in disentangled layouts, providing a global action space on LSTM for visual navigation. VGM (Kwon et al., 2021) presents visual graph memory based on GCN, which includes unsupervised image representations for navigation history. In the realm of image-goal navigation, memory-augmented reinforcement learning (Mezghani et al., 2022) integrates an external memory mechanism with representations of past observations into the navigation policy. MemoNav (Li et al., 2024a) introduces a memory model based on GCN and LSTM, attending to short- and long-term memory while efficiently managing memory by forgetting information below a threshold. SMT (Fang et al., 2019) incorporates attention mechanisms to exploit spatio-temporal dependencies, maintaining long time horizons for navigation. Furthermore, MultiON (Wani et al., 2020) proposes navigation tasks to test agents' ability to locate previously observed goal objects. In visual odometry, memory mechanisms also enhance performance. The Deep Visual Odometry With Adaptive Memory model (Xue et al., 2022) employs selective memory based on RNNs to improve visual odometry accuracy and adaptability. For manipulation reasoning and planning, RDMemory (Huang et al., 2024b) encodes object-oriented memory into a multi-object manipulation framework based on transformers, facilitating sophisticated reasoning and planning capabilities.

### 5.3.3 Application

Memory is a critical component in the evolution of multimodal embodied agents, enabling them to seamlessly integrate and process diverse inputs such as visual, auditory, and textual data for adaptive, context-aware decision-making. Recent advancements highlight the role of memory-enhanced agents in tasks like autonomous navigation, healthcare assistance, interactive education, and smart home systems. By leveraging memory, these agents can retain past interactions, learn from experiences, and adapt to complex, dynamic environments, significantly enhancing their ability to understand, plan, and execute tasks across domains. Applications range from disaster response robots that utilize spatial memory for efficient navigation to personalized assistants that adapt based on user preferences and history. Memory's role in providing continuity and context allows these agents to go beyond static task execution, achieving higher levels of intelligence and functionality in both real-world and virtual scenarios.

### 5.4 Limitations, Open Questions, Discussion

While existing research has achieved significant progress in long-sequence multimodal tasks—such as long video understanding and long document processing—horizontal scaling challenges remain particularly pronounced in multimodal systems. This stems from the inherent abundance of visual tokens generated by patch-based image processing methods. Two critical factors exacerbate these challenges:

- Multimodal interaction inherently demands multi-turn reasoning, necessitating robust long-term memory to retain contextual coherence.

- Time-series modalities (e.g., audio, video, or streaming data) require long-term memory retention to model temporal dependencies effectively.

- Embodied learning requires memorizing multimodal information from the interaction between the agent and the real world.

Addressing these memory challenges—balancing computational efficiency with model effectiveness—represents a pivotal frontier for advancing multimodal systems.

## 6 Conclusion

In this report, we present a narrative review of three distinct types of memory integrated into large language models (LLMs): implicit memory, which is embedded within model parameters; explicit memory, which involves external storage and retrieval mechanisms; and agent memory, which captures persistent interactions with environments. Additionally, we systematically examine memory mechanisms specifically designed for and utilized by multimodal LLMs. Our survey is meticulously structured to trace the developmental trajectory of memory mechanisms, spanning from foundational concepts to the most recent advancements. We provide not only a comprehensive map of the landscape of LLM memory but also detailed explorations of pivotal milestones, rigorous empirical evaluations, and a forward-looking vision for the field's future development.

## 7 Future Work and Limitations

Based on this comprehensive review, we outline several key considerations and future research directions. First, there is a critical need to advance our understanding of the internal mechanisms of Transformer architectures and to develop more effective frameworks for implicit memory modeling. Second, enhancing the long-context processing capabilities of LLMs, either through extended context windows or retrieval-augmented generation (RAG), is essential; however, each approach presents trade-offs in terms of computational efficiency and scalability. Third, dynamic memory adaptation, inspired by human learning strategies such as recursive retrieval and experience reflection, holds promise for improving reasoning and communication in agent-based systems. Finally, multimodal systems face particular challenges stemming from the high volume of visual tokens, the complexity of multi-turn reasoning, and the temporal dependencies inherent in time-series data. Addressing these challenges calls for the development of scalable, memory-efficient architectures that support coherent, adaptive, and multimodal long-term learning.

In our survey, we discuss memory in LLMs, agents, and multimodal LLMs, but we do not provide a unified evaluation framework for all memory types due to the diverse aspects each memory mechanism emphasizes. Moreover, we do not propose a single platform or system that integrates these various memory mechanisms. We invite fellow researchers to engage with our findings and collaborate toward the advancement of memory-augmented AI systems.

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
