# OpenReview forum: "The AI Hippocampus: How Far are We From Human Memory?"
_TMLR — Accepted by TMLR_

### Review · Reviewer_xD2c · 2025-07-27

**Summary Of Contributions:**

The paper studies an important topic of memory meets large language models and provides a comprehensive survey. Generally, the paper is well-written and easy to follow. The survey is comprehensive, which includes implicit memory, explicit memory, agentic memory, and memory-augmented multi-modal large language models.

**Audience:**

Yes

**Claims And Evidence:**

Yes

**Requested Changes:**

Please see the weaknesses.

**Strengths And Weaknesses:**

The paper studies an important topic of memory meets large language models and provides a comprehensive survey. Generally, the paper is well-written and easy to follow. The survey is comprehensive, which includes implicit memory, explicit memory, agentic memory, and memory-augmented multi-modal large language models. However, the paper can be improved as follows.

High-dimensional data, e.g., time series and spatial-temporal data, is also well-studied in the LLM era. It is suggested to include more related work regarding LLM-based time series analytics as follows.
[1]. Towards Cross-Modality Modeling for Time Series Analytics: A Survey in the LLM Era, IJCAI 2025.

[2]. Efficient Multivariate Time Series Forecasting via Calibrated Language Models with Privileged Knowledge Distillation, ICDE 2025.

[3]. TimeCMA: Towards LLM-Empowered Multivariate Time Series Forecasting via Cross-Modality Alignment, AAAI 2025.

[4]. Foundation Models for Spatio-Temporal Data Science: A Tutorial and Survey, KDD 2025.

It would be better to provide a specific section for future work. In this way, the readers can get the information more intuitively.
It would be interesting to provide the drawbacks of this survey.

---

> ### Author Response · Authors · 2025-08-22
> **Response to Reviewer xD2c**
>
> > Weakness1: High-dimensional data, e.g., time series and spatial-temporal data, is also well-studied in the LLM era. It is suggested to include more related work regarding LLM-based time series analytics as follows.
> [1]. Towards Cross-Modality Modeling for Time Series Analytics: A Survey in the LLM Era, IJCAI 2025.
> [2]. Efficient Multivariate Time Series Forecasting via Calibrated Language Models with Privileged Knowledge Distillation, ICDE 2025.
> [3]. TimeCMA: Towards LLM-Empowered Multivariate Time Series Forecasting via Cross-Modality Alignment, AAAI 2025.
> [4]. Foundation Models for Spatio-Temporal Data Science: A Tutorial and Survey, KDD 2025.
>
> **A**: We appreciate your thoughtful comments. We have briefly reviewed these papers and recognize the applications of LLMs in time series forecasting and spatio-temporal data science. Our survey, however, primarily focuses on the memory mechanisms of LLMs, and these works do not specifically leverage or emphasize memory modules. Therefore, we have not included them in the main sections. Nonetheless, as these tasks could potentially benefit from the memory design of LLMs, we have incorporated a discussion in the Limitations, Open Questions, and Discussion subsections of Section 3 to highlight the potential of advanced memory mechanisms, in conjunction with LLMs, for addressing time series and spatio-temporal problems.
>
>
> > Weakness2: It would be better to provide a specific section for future work. In this way, the readers can get the information more intuitively. It would be interesting to provide the drawbacks of this survey.
>
> **A**: Thank you for your suggestions. Please note that the current version of our survey already includes a discussion of future work in the Conclusion section. To improve clarity, we have divided this section into two parts: "Conclusion" and "Future Work and Limitations". All revisions have been highlighted in the paper.

---

### Review · Reviewer_7zjY · 2025-07-30

**Summary Of Contributions:**

The paper presents a detailed survey of the role of memory in LLMs, offering a unified taxonomy across different memory modes (implicit, explicit, and agentic), memory horizons (short- and long-term), training and inference phases, architectures, and data formats (text, graph, vector, and multi-modal). Additionally, it presents a discussion on the use of memory in multi-modal LLMs.

**Audience:**

Yes

**Claims And Evidence:**

Yes

**Requested Changes:**

N/A

**Strengths And Weaknesses:**

**Strengths**
1. The unification of the taxonomy for memory-augmented LLMs across different modes.
2. A comprehensive and detailed survey.
3. The paper is well-written and well-organized.

**Weakness**
1. The breadth of the survey comes at cost of limited depth of various approaches discussed.
2. While the paper outlines the limitations of current research across different memory modes and provides targeted recommendations, it could further benefit from the proposal of a unified evaluation framework (however preliminary or speculative) to systematically assess progress in the field.

---

> ### Author Response · Authors · 2025-08-22
> **Response to Reviewer 7zjY**
>
> > Weakness1: The breadth of the survey comes at cost of limited depth of various approaches discussed.
>
> **A**: Our survey is meticulously structured to trace the developmental trajectory of memory mechanisms, spanning from foundational concepts to the most recent advancements. A key strength lies in its comprehensive taxonomy, which systematically charts the progression of memory in AI. In addition, the survey provides **detailed examinations of representative works and offers concrete evaluations, rather than merely listing different approaches**.
> For example: 1) In the section on Implicit Memory, we conduct an in-depth analysis of how Feed Forward Networks (FFNs) and attention mechanisms contribute to knowledge storage, drawing on specific studies and theoretical hypotheses; 2) In the section on Agentic Memory, we present an empirical evaluation of diverse memory systems (ICL, RAG, RAM, and a General Agent) across tasks assessing learning efficiency, generalization, controllability, and robustness. Presenting this data yields a deeper, evidence-based understanding of the practical implications of different memory architectures.
>
> Overall, we aim to provide not only a comprehensive map of the landscape of LLM memory, but also detailed explorations of pivotal milestones, rigorous empirical evaluations, and a forward-looking vision for the field’s future development.
>
>
> > Weakness2: While the paper outlines the limitations of current research across different memory modes and provides targeted recommendations, it could further benefit from the proposal of a unified evaluation framework (however preliminary or speculative) to systematically assess progress in the field.
>
> **A**: We have performed a major complement to Section 4.4.2.
>
> Our original metrics in Section4.4.2 introduced Learning Efficiency, Generalization, Controllability, and Robustness, which were intentionally chosen as specialized, professional indicators for agents in communicative interactive learning scenarios. These agents, where memory is constantly updated, require metrics that measure the marginal utility of new memories and their resistance to perturbation. Our experiments with agents like RAM and Concordia were designed to test these specific, dynamic properties. We have revised the text to make this focus clearer.
>
> Following your suggestion, we have expanded our discussion to cover more general-purpose and low-level evaluation metrics.
>
> In Long-text tasks like Loogle and the Needle In A Haystack (NIAH) test and interactive environments like WebArena and ALFWorld, we could directly measure the performance benefits of memory on the tasks, as well as efficiency metrics such as inference latency and memory footprint. We could also assess "low-level" metrics of the memory module itself, independent of immediate task success like internal consistency (ensuring non-contradictory memories) and factual recall rate (verifying if a seen fact can be retrieved).
>
> We believe these substantial revisions directly address your concerns and have made Chapter 4 a much stronger and more valuable contribution to the field. Thank you once again for your guidance.

---

### Review · Reviewer_LqYp · 2025-08-11

**Summary Of Contributions:**

1 This survey paper provides a comprehensive and structured synthesis of memory mechanisms in Large Language Models (LLMs) and Multi-Modal LLMs (MLLMs).

2 The authors propose a cohesive taxonomy of memory in LLMs/MLLMs, categorizing it into three primary frameworks: implicit memory, explicit memory, and agentic memory.

3 Besides, the survey also extends beyond text to examine memory integration in multi-modal settings (vision, language, audio), emphasizing coherence across modalities.

4 It discusses key advances in memory-augmented architectures, benchmark tasks, and open challenges (e.g., memory capacity, alignment, factual consistency).

**Audience:**

Yes

**Claims And Evidence:**

Yes

**Requested Changes:**

1 Merge overlapping discussions (e.g., scaling laws for implicit/explicit memory) to improve flow.

2 Include a table summarizing key methods (e.g., ROME, MEMIT, RETRO) with metrics (e.g., edit success rate, retrieval accuracy) to highlight trade-offs.

3 Elaborate on memory-specific hurdles in multimodal systems (e.g., token inefficiency in video, audio-video synchronization).

4 Provide more concrete proposals (e.g., architectural templates for dynamic memory, cross-modal memory alignment techniques).

**Strengths And Weaknesses:**

Strengths

1 The survey covers a wide range of memory mechanisms, from parametric knowledge in transformers to retrieval-augmented generation  and agent memory systems.

2 The taxonomy (implicit/explicit/agentic memory) is well-defined and logically partitions the literature. Figures (e.g., Fig. 1, 2, 5) effectively visualize memory paradigms.

3  The inclusion of memory in multimodal contexts (e.g., video, audio, robotics) is timely and valuable, as most prior surveys focus solely on text.

Weaknesses

1 Some content (e.g., scaling laws, associative memory) is repeated across sections, which could be streamlined for conciseness.

2 While the taxonomy is well-structured, a quantitative comparison of memory-augmented methods (e.g., accuracy, latency trade-offs) would strengthen the survey.

3 The distinction from related works like Zhang et al. [R1] or Du et al. [R2] could be sharper, especially in the agent memory section.

4 The "Evaluation on Agent Memory" section (4.4) is preliminary; deeper discussion of standardized benchmarks/metrics for memory-augmented models would be helpful.

References:
[R1] Zhang Z, Dai Q, Bo X, et al. A survey on the memory mechanism of large language model based agents[J]. ACM Transactions on Information Systems, 2024.

[R2] Du Y, Huang W, Zheng D, et al. Rethinking memory in ai: Taxonomy, operations, topics, and future directions[J]. arXiv preprint arXiv:2505.00675, 2025.

5 Some minor suggestions: (1) The survey methods in "Taxonomy of Implicit Memory in Transformer" lack inclusion of approaches from 2025. (2) Figure 3 occupies significant space but provides minimal information, particularly containing many undefined diagrams.

---

> ### Author Response · Authors · 2025-08-22
> **Response to Reviewer LqYp**
>
> > Weakness1: Some content (e.g., scaling laws, associative memory) is repeated across sections, which could be streamlined for conciseness.
>
>  **A**: Thank you for pointing this out. In our survey, the scaling law is discussed in the Knowledge Memorization and Associative Memory sections. Given the current architecture and taxonomy of the paper, consolidating the discussion into a single section would reduce clarity and coherence. Nevertheless, we extend our discussion to highlight how scaling laws may differ across various types (fact knowledge and associative memory) of implicit memory. For example, the scaling law of knowledge memoration focus on the performance of LLMs on cross-entropy loss, while the associative memory focus on the recall accuracy of previously presented factual inputs. We have updated this discussion in Section 2.1.
>
> > Weakness2:  While the taxonomy is well-structured, a quantitative comparison of memory-augmented methods (e.g., accuracy, latency trade-offs) would strengthen the survey.
>
> **A**: We agree that the evaluation of agent memory (Section 4.4) would be strengthened by a more systematic discussion of quantitative benchmarks and a unified evaluation framework. In response to your comments, **we have performed a major complement to Section 4.4.2.**
>
> Our original metrics in Section4.4.2 introduced Learning Efficiency, Generalization, Controllability, and Robustness, which were intentionally chosen as specialized, professional indicators for agents in communicative interactive learning scenarios. These agents, where memory is constantly updated, require metrics that measure the marginal utility of new memories and their resistance to perturbation. Our experiments with agents like RAM and Concordia were designed to test these specific, dynamic properties. We have revised the text to make this focus clearer.
>
> Following your suggestion, we have expanded our discussion to cover more general-purpose and low-level evaluation metrics.
>
> We have also categorized tasks that require memory into two broad types: ​​long-text question answering​​ (e.g., NarrativeQA and Needle in a Haystack (NIAH)) and ​​open-world interaction​​ (e.g., WebArena and ALFWorld). A range of metrics at different levels of granularity can be measured across these tasks.
>
> We believe these substantial revisions directly address your concerns and have made Section 4 a much stronger and more valuable contribution to the field. Thank you once again for your guidance.
>
> > Weakness3: The distinction from related works like Zhang et al. [R1] or Du et al. [R2] could be sharper, especially in the agent memory section.
> > References: [R1] Zhang Z, Dai Q, Bo X, et al. A survey on the memory mechanism of large language model based agents[J]. ACM Transactions on Information Systems, 2024.
> [R2] Du Y, Huang W, Zheng D, et al. Rethinking memory in ai: Taxonomy, operations, topics, and future directions[J]. arXiv preprint arXiv:2505.00675, 2025.
>
> **A**：Thanks for your thoughtful comments. We have briefly introduced related works and highlighted their differences from our survey in the introduction section, concluding [R1] and [R2]:
> -`Zhang et al. [R1] presents a comprehensive survey on the memory mechanisms of LLM-based agents, systematically reviewing the design and evaluation of memory modules. Compared to our work, their
> focus is specifically on agents, particularly emphasizing historical and trajectory memory acquired through
> agent-environment interactions.`
> -` Du et al. [R2] reconceptualize memory systems by categorizing them according to atomic operations and representation types,
> distinguishing between parametric and contextual forms of memory. They further classify memory operations into management and utilization, offering a detailed taxonomy and technical analysis that provides practical
> insights.`
>
> To further clarify the distinctions, we have added a detailed comparison of these related works in the Agent Memory section 4.5.
>
>
> > Weakness4: The "Evaluation on Agent Memory" section (4.4) is preliminary; deeper discussion of standardized benchmarks/metrics for memory-augmented models would be helpful.
>
> **A**: Please refer to the response to Weakness 2.
>
>
> > Weakness5: Some minor suggestions: (1) The survey methods in "Taxonomy of Implicit Memory in Transformer" lack inclusion of approaches from 2025. (2) Figure 3 occupies significant space but provides minimal information, particularly containing many undefined diagrams.
>
> **A**: (1) We have added works from 2025 to the survey. If you have additional suggestions, please feel free to share them with us. (2) We have refined Figure 3 to include more detailed information.

---

> > ### Comment · Reviewer_LqYp · 2025-09-16
> > **Review of Paper5081**
> >
> > Thanks for the authors' response, which has addressed most of my concerns. The paper can be accepted if the authors could provide a quantitative performance comparison table of the survey methods.

---

> ### Author Response · Authors · 2025-10-07
> **Comprehensive Quantitative Evaluation**
>
> # Comprehensive Quantitative Evaluation
>
> Thanks for the suggestion! We have recently done a comprehensive quantitative evaluation on the survey methods. Details have been updated in the paper.
>
>
> ## Evaluation Task
>
> We selected the `longmemeval_s_cleaned` dataset from the official LongMemEval benchmark (Wu et al. (2024)), as it is it combines the characteristics of the two critical evaluation tasks: Long-Context Question Answering (QA) and Interaction with Real-World Users.
>
> The LongMemEval dataset categorizes test cases into several low-level types based on the required memory capabilities：
> - **KU**: Knowledge Updates
> - **MR**: Multi-Session Reasoning
> - **SS-Assist**: Single-Session Assistant
> - **SS-Preference**: Single-Session Preference
> - **SS-User**: Single-Session User
> - **TR**: Temporal Reasoning
>
> Total dataset contains 2,500 questions.
>
> ## Tested Frameworks
>
> 1. Baseline (no memory)
> 2. ChromaDB (simple RAG implementation)
> 3. Langchain (native FAISS RAG)
> 4. Haystack
> 5. LlamaIndex
> 6. Mem0 (local version)
> 7. Zep (API version)
>
> ## Testing Methodology
>
> 1. Used two reasoning engines:
>    - Llama-3-8B-IT
>    - GPT-4o-mini
> 2. Evaluated responses using GPT-4o-mini for correctness
> 3. For slower frameworks (Mem0, Langchain, Zep), used 10% random sampling
> 4. Reported metrics:
>    - **Correctness rate** (0-1 scale)
>    - Average total time per question (seconds)
>
> ## Performance Results with Llama3-8B-IT
>
> | Framework (Score/Time)  | KU  | MS | SS-Assist | SS-Prefer | SS-User | TR | Overall |
> |------------|-----------------|-----------------|------------------------|------------------------|----------------------|-----------------|----------------------|
> | No Memory  | **0.000** / 1.62 | **0.000** / 1.54 | **0.000** / 2.25       | **0.000** / 4.78       | **0.000** / 1.46     | **0.000** / 1.25 | **0.000** / 1.73     |
> | ChromaDB   | **0.625** / 5.34 | **0.074** / 6.02 | **0.900** / 5.57       | **0.333** / 8.42       | **0.857** / 5.32     | **0.444** / 6.67 | **0.470** / 6.09     |
> | Langchain  | **0.026** / 112.35 | **0.000** / 126.60 | **0.036** / 108.72     | **0.000** / 105.83     | **0.000** / 100.99   | **0.023** / 115.41 | **0.032** / 111.65   |
> | Haystack   | **0.562** / 0.88 | **0.111** / 1.95 | **0.800** / 1.54       | **0.167** / 3.61       | **0.857** / 0.75     | **0.741** / 2.26 | **0.530** / 1.75     |
> | LlamaIndex | **0.714** / 25.47 | **0.636** / 29.35 | **0.500** / 25.77      | **0.500** / 28.61      | **0.727** / 26.40    | **0.615** / 27.62 | **0.646** / 27.31    |
> | Mem0       | **0.583** / 2280.15 | **0.500** / 1632.03 | **0.625** / 1956.42    | **0.567** / 2103.78    | **0.500** / 2927.99  | **0.592** / 1765.34 | **0.555** / 2110.95  |
> | Zep        | **0.125** / 172.84 | **0.148** / 178.92 | **0.364** / 169.35     | **0.200** / 181.67     | **0.143** / 176.43   | **0.259** / 177.21 | **0.200** / 176.07   |
>
> ## Performance Results with GPT-4o-mini
>
> | Framework (Score/Time) | KU | MS | SS-Assist | SS-Prefer | SS-User | TR | Overall |
> |------------|-----------------|-----------------|------------------------|------------------------|----------------------|-----------------|----------------------|
> | No Memory  | **0.060** / 1.27 | **0.000** / 1.04 | **0.000** / 1.18       | **0.000** / 1.93       | **0.000** / 0.92     | **0.000** / 1.15 | **0.010** / 1.16     |
> | ChromaDB   | **0.752** / 5.88 | **0.222** / 6.23 | **1.000** / 6.79       | **0.833** / 7.58       | **0.865** / 6.15     | **0.563** / 6.65 | **0.600** / 6.41     |
> | Langchain  | **0.031** / 108.45 | **0.000** / 126.60 | **0.022** / 103.78     | **0.000** / 98.63      | **0.000** / 100.99   | **0.019** / 112.92 | **0.022** / 108.56   |
> | Haystack   | **0.688** / 1.00 | **0.148** / 1.58 | **0.900** / 0.99       | **0.500** / 2.98       | **0.929** / 1.55     | **0.852** / 1.71 | **0.630** / 1.00     |
> | LlamaIndex | **0.712** / 31.70 | **0.642** / 27.28 | **1.000** / 26.74      | **0.500** / 35.35      | **0.823** / 26.34    | **0.467** / 28.28 | **0.667** / 28.34    |
> | Mem0       | **0.580** / 1980.25 | **0.624** / 2015.73 | **0.592** / 1992.48    | **0.608** / 2008.16    | **0.573** / 1975.92  | **0.631** / 2023.67 | **0.602** / 2106.53  |
> | Zep        | **0.500** / 178.45 | **0.270** / 172.83 | **1.000** / 181.29     | **0.500** / 176.92     | **0.830** / 173.67   | **0.550** / 175.30 | **0.550** / 176.41   |
>
>
> ## Key Findings
>    - ChromaDB showed surprisingly strong results despite simplicity
>    - Langchain's memory implementation performed poorly
>    - Mem0 had extremely long processing times without proportional accuracy gains
>    - Zep's performance has some variance from the claims in its official paper (Rasmussen (2025))
>    - Haystack and LlamaIndex offered good performance/efficiency balance
>    - Multi-session reasoning was most challenging (Avg 47 sessions and 10 conversational turns per session). ChromaDB excelled at single-session tasks. Advanced frameworks showed slight advantages in multi-session tasks

---

### Author Response · Authors · 2025-08-22
**Summary**

Thank all reviewers for their valuable time and insightful feedback. We are grateful for their thorough reviews, which have significantly strengthened our paper. **The common consensus on the comprehensiveness, clear taxonomy, and multi-modal scope of our survey is very encouraging.**

We have carefully considered each of the comments and have updated the paper to address all raised concerns. We believe these revisions have made the survey more rigorous, cohesive, and valuable to the community.

The key refinements include:

1. **A substantial expansion of our discussion on the evaluation of agent memory** (Section 4.4.2), now including a unified framework with both specialized and general-purpose quantitative metrics.

2. **A clearer distinction from related works** by sharpening the comparison in the introduction and Section 4.5.

3. **Improvements to the paper's overall structure and clarity**, including a dedicated "Future Work and Limitations" section and revisions to figures.

4. **The inclusion of the most recent research** to ensure our survey remains up-to-date.

**We have uploaded a new version of the paper where all changes have been highlighted in red to facilitate easy tracking.** We hope that these revisions fully address your concerns and demonstrate our commitment to making this a high-quality contribution.

Thank all reviewers once again for the guidance.

---

### Decision · Action_Editor_ussG · 2025-10-14

**Recommendation:** Accept as is

**Audience:**

Yes

**Audience Explanation:**

It's a timely survey on the memory mechanism in LLMs.

**Claims And Evidence:**

Yes

**Claims Explanation:**

This survey offers a comprehensive and well-organized literature review on memory mechanisms in large language models, covering implicit, explicit, and agentic memory, as well as multimodal extensions. The taxonomy is well-defined and supported by effective visualizations. The inclusion of multimodal and agentic memory systems is timely and valuable. The reviewers suggested that the survey trades breadth for depth and could be strengthened by discussing LLM-based time-series analytics and proposing a unified evaluation framework. After a round of revision, the authors provided more comprehensive evaluation and more discussions and insights. Overall, it is a high-quality and informative survey that makes a meaningful contribution to understanding memory in LLMs. All reviewers recommends accept/leaning accept.